# Multivariate bias corrections of climate simulations: Which benefits for which losses?

Bastien François[1], Mathieu Vrac[1], Alex J. Cannon[2], Yoann Robin[3], and Denis Allard[4]

[1]Laboratoire des Sciences du Climat et l'Environnement (LSCE-IPSL) CNRS/CEA/UVSQ, UMR8212, Université Paris-Saclay, Gif-sur-Yvette, France
[2]Climate Research Division, Environment and Climate Change Canada, Victoria, BC, Canada
[3]Centre National de Recherches Météorologiques, Université de Toulouse, CNRS, Météo-France, Toulouse, France
[4]INRAE, BioSP, 84914, Avignon, France

**Correspondence:** B. François (bastien.francois@lsce.ipsl.fr)

**Abstract.** Climate models are the major tools to study the climate system and its evolutions in the future. However, climate simulations often present statistical biases and have to be corrected against observations before being used in impact assessments. Several bias correction (BC) methods have therefore been developed in the literature over the last two decades, in order to adjust simulations according to historical records and obtain climate projections with appropriate statistical attributes. Most of the existing and popular BC methods are univariate, i.e., correcting one physical variable and one location at a time, and thus can fail to reconstruct inter-variable, spatial or temporal dependencies of the observations. These remaining biases in the correction can then affect the subsequent analyses. This has led to further research on multivariate aspects for statistical postprocessing BC methods. Recently, some multivariate bias correction (MBC) methods have been proposed, with different approaches to restore multidimensional dependencies. However, these methods are not yet fully apprehended by researchers and practitioners due to differences in their applicability and assumptions, therefore leading potentially to different results. This study is intended to intercompare four existing MBCs to provide end-users with aid in choosing such methods for their applications. For evaluation and illustration purposes, these methods are applied to correct simulation outputs from one climate model through a cross-validation method, which allows for the assessment of inter-variable, spatial and temporal criteria. Then, a second cross-validation method is performed for assessing the ability of the MBC methods to account for the multi-dimensional evolutions of the climate model. Additionally, two reference datasets are used to assess the influence of their spatial resolution on (M)BC results. Most of the methods reasonably correct inter-variable and inter-site correlations. However, none of them adjust correctly the temporal structure as they generate bias corrected data with usually weak temporal dependencies compared to observations. Major differences are found concerning the applicability and stability of the methods in high-dimensional contexts, and in their capability to reproduce the multi-dimensional changes of the model. Based on these conclusions, perspectives for MBC developments are suggested, such as methods to adjust not only multivariate correlations but also temporal structures and allowing to account for multi-dimensional evolutions of the model in the correction.

# 1 Introduction

Representing precisely the climate system and the interactions between its components is a major challenge not only for climate
modellers but also for scientists working on impact, mitigation and adaptation issues relating to Climate Change. Indeed, it
is now common that climate change impact studies, e.g., in hydrology, environmental science or economics, use Global and
Regional climate model (GCM and RCM) simulations as inputs into impact models. However, in spite of continued scientific
progress in climate modelling, climate simulations often remain biased compared to observations (Christensen et al., 2008).
This means that their statistical attributes such as mean, variance, extreme, or even dependence structures between several
variables and/or sites can differ from those calculated based on historical records. Therefore, using plain simulations can
significantly affect the results of impact studies.

To solve this issue, many statistical bias correction (BC) methods have been developed, in order to correct the statistical
discrepancies of the simulations before climate change assessment studies. Most of the BC methods in use are designed to
adjust univariate distribution features of climate variables, such as the mean (e.g., Delta method, Xu, 1999), the variance
(e.g., simple scaling adjustment, Berg et al., 2012) or quantiles (e.g., "quantile-mapping", Haddad and Rosenfeld, 1997). The
latter technique received a notable success, since it permits to adjust the mean, the variance and any quantile of the climate
variables. Its theoretical framework has been conducive to the development of multiple versions of quantile-based methods
(e.g., Panofsky and Brier, 1958; Déqué, 2007; Gudmundsson et al., 2012; Vrac et al., 2012). However, all these univariate
BC methods are designed to correct variables independently, i.e. are applied separately for each physical variable at each
specific location (e.g., grid cell). Although univariate distribution features are adjusted according to references, it can generate
inappropriate multivariate situations where the dependence structure between variables and sites is not corrected from the
model and misrepresented (Maraun, 2013), or even modified. Ignoring the observed inter-variable and inter-site dependencies
in the correction procedure can result in obtaining corrected outputs with inappropriate physical laws, and thereby distorting the
results of impact studies (Zscheischler et al., 2019). It is therefore of paramount importance to adjust the dependence structures
of climate simulations, in addition to 1d-characteristics, before using it in subsequent studies.

These methodological issues have led up to the recent development of a few multivariate bias correction (MBC) methods.
Not only do these methods adjust univariate distribution features, they are also aimed at correcting the dependence structure
of climate simulations. Recent studies have shown that univariate BC methods can already provide adequate results for certain
specific regional impact studies (Yang et al., 2015; Casanueva et al., 2018), and that using MBC methods does not neces-
sarily present substantial benefits (Räty et al., 2018). However, this does not call into question the interest of MBC methods
as these specific results cannot be generalized to each method and application. In particular, MBC methods could be valu-
able in larger-scale impact modelling frameworks such as compound events, where the combination of physical processes
across multiple spatial and temporal scales leads to significant impacts (Zscheischler et al., 2018). As mentioned by Vrac
(2018), and completed by Robin et al. (2019), MBC methods may be grouped into three main categories of approaches: the
"marginal/dependence" correction approach, the "successive conditional" correction approach, and the "all-in-one" correction
approach. The "marginal/dependence" category is made up of multivariate bias adjustment methods correcting separately the

marginal distributions and the dependence relationships of climate simulations (e.g., Bárdossy and Pegram, 2012; Mehrotra and Sharma, 2016; Vrac, 2018; Nahar et al., 2018; Cannon, 2018). In the "all-in-one" category, multivariate BC methods correct the 1d-marginal properties and dependence structures altogether at the same time (e.g., Robin et al., 2019). At last, "successive conditional" MBC methods perform successive corrections, conditionally on the variables already corrected (e.g., Bárdossy and Pegram, 2012; Dekens et al., 2017). In particular, this latter category has two major limitations. First, the quality of the correction can change depending on the ordering of the variables to correct (see, e.g., Piani and Haerter, 2012). Second, the number of variables already corrected increases at each iteration step, which progressively reduces the number of data available for the correction, making it less and less robust. Accordingly, these methodological limits call into question the applicability of "successive conditional" BC methods for multivariate bias adjustment of high-dimensional climate simulations.

Additionally to the methodological distinction described above, the few existing multivariate BC methods are based on the use of different statistical techniques. They may also present differences in terms of assumptions and philosophical features, e.g. deterministic versus stochastic. Consequently, the quality of the correction outputs can vary largely from one method to another, depending on their characteristics. It is hence crucial, in particular for end-users, to carefully evaluate the suitability of these multivariate BC methods, and identify their advantages and limits, not only between the different categories of methodological approaches but also between the different statistical techniques and assumptions. In this study, we present an analysis of four multivariate BC methods, and assess their performances in terms of adjustment of dependence structures for temperature and precipitation time series. We focus in particular our intercomparison on methods belonging to the "marginal/dependence" and the "all-in-one" categories. Due to the previously mentioned limitations of the "successive conditional" approach, no methods belonging to this category are investigated. The selected four MBC methods present differences in terms of conceptual features, statistical techniques used and assumptions. In particular, MBCs with different assumptions on non-stationarity are selected, i.e. differing on how they consider the simulated multi-dimensional changes between present (i.e calibration) and future (i.e. projection) periods in the correction procedure. Moreover, in order to assess the potential benefits of using multivariate BC methods relatively to univariate ones, one univariate quantile-mapping based BC method is included in the study as a benchmark. It provides a more extensive intercomparison framework in which quality of BC outputs can be assessed and compared by evaluating univariate, inter-variable, spatial, and temporal properties, as well as multi-dimensional changes.

In addition, each BC method is applied to correct climate model outputs over France and three sub-regions according to two distinct reference datasets with different spatial resolutions. This permits to assess the potential influence of the reference spatial resolution on bias correction results and to delineate guidance on relevant good practices for end-users concerning this aspect.

This paper is organised as follows: Section 2 describes the model and reference data used and Section 3 presents the BC methods intercompared. Then, Section 4 presents the experimental set-up used in this study, while Section 5 displays the results of the intercomparison. Finally, our findings are summarised, discussions are given and perspectives for future research are proposed in Section 6.

## 2 Model and reference data

Institut Pierre-Simon Laplace (IPSL) coupled model (Marti et al., 2010; Dufresne et al., 2013) daily data with a 1.25° by 2.5° spatial resolution are used in this study as model data to be corrected. Simulations of the scenario of atmospheric $CO_2$ concentration pathway associated with a radiative forcing of $+8.5$ W.m$^{-2}$ (RCP 8.5 scenario, i.e. the scenario with highest $CO_2$ concentration) are selected. Daily temperature (T2) and precipitation (PR) time series from 1 January 1979 to 31 December 2016 are extracted over the geographical area of France ([-5° E; 10° E] $\times$ [42° N; 51° N]), which corresponds to 321 continental grid cells.

As BC methods require a reference dataset to adjust the simulations, daily temperature and precipitation time series with a 0.5° by 0.5° spatial resolution are first used from the "WATCH Forcing Data methodology applied to ERA-Interim data" (WFDEI) from the EU WATCH project (Weedon et al., 2014) over the same geographical area of France. Note that, as spatial resolution between WFDEI and IPSL-CM5 are different, IPSL model data are regridded by a nearest neighbour technique to associate each IPSL grid cell to its nearest WFDEI grid cell center. Hence, in the following, the IPSL data will be used at the 0.5° spatial resolution corresponding to that of the WFDEI reference dataset.

To assess the influence of the reference spatial resolution on BC results, we use another reference gridded dataset for France with finer resolution: the "Systeme d'Analyze Fournissant des Renseignements Atmosphériques à la Neige" (SAFRAN) re-analysis dataset (Vidal et al., 2010). Daily T2 and PR time series from SAFRAN have a 8km$\times$8km spatial resolution and divide France into 8981 continental grid cells. IPSL data are regridded to the 8km$\times$8km SAFRAN resolution using nearest neighbour technique. Once regridded IPSL simulations are obtained, each MBC method can be applied. However, as some MBC algorithms have difficulties in practice in very high-dimensional context (here for 8981 grid cells), we restrict the application of MBCs with SAFRAN reference dataset over the Brittany region, located in the northwest part of France ([-5°E ; 2°E ] $\times$ [ 47°N ; 49°N]), which corresponds to 345 continental grid cells. Note that we selected this sub-region of Brittany for SAFRAN, i.e. at fine resolution, in order to have a similar number of grid cells as for France selected with the WFDEI reference dataset, i.e. at coarser resolution. MBC methods have also been applied and evaluated over two others sub-regions of 345 grid cells located respectively around the Paris area and in southeast France. For sake of clarity, as same results were obtained for each of the sub-regions, we will only present the results for Brittany in the rest of this study.

## 3 Multivariate bias correction methods

This section presents a brief description of the univariate BC method and the four multivariate BC methods implemented in this study. As a reminder, results from the univariate CDF-t method serve as a benchmark to measure the benefits of considering multivariate aspects in the correction procedure instead of using univariate BC methods. For sake of clarity, Table 1 provides a concise summary of the different attributes that make the BC methods distinct.

## 3.1 "Cumulative Distribution Function - Transform" (CDF-t)

The "Cumulative Distribution Function - Transform" (CDF-t) method is a univariate BC method initially proposed by Michelangeli et al. (2009) to correct the univariate distribution of a modeled climate variable. Since then, CDF-t has been applied for various studies (e.g., Tramblay et al., 2013; Tobin et al., 2015; Defrance et al., 2017; Famien et al., 2018; Guo et al., 2018) and specific variants have been developed (e.g., Kallache et al., 2011; Vrac et al., 2016). The CDF-t approach applies, independently to each variable, a univariate transfer function $T$, which permits to link the cumulative distribution function (CDF) of a variable of interest in the model simulations to that of the reference dataset. By assuming that $T$ is valid in a climate different from that of the calibration period, a new CDF for the bias corrected variable over the projection period is generated. Then, a quantile-quantile approach is performed between the new (reference) CDF and the CDF from the model simulations during the projection period to derive bias-corrected data. This two-step procedure permits to take into account potential changes (between calibration and projection periods) of the univariate distribution in the correction procedure. For the special case of precipitation, the "Singularity Stochastic Removal" version of CDF-t (Vrac et al., 2016) is applied to correct both precipitation occurrences and intensities. More details about CDF-t can be found in Appendix A. In the following subsections, the four MBC methods are presented.

## 3.2 "Rank Resampling for Distributions and Dependences" ($R^2D^2$)

The "Rank Resampling for Distributions and Dependences" ($R^2D^2$) method, developed by Vrac (2018) in the context of "marginal/dependence" category, is an extension of the "Empirical Copula - Bias Correction" (EC-BC, Vrac and Friederichs, 2015) method. $R^2D^2$ is based on a reordering technique named the Schaake Shuffle. Originally described by Dr. J. Schaake in 2002, it was introduced in the scientific literature by Clark et al. (2004) to postprocess temperature and precipitation forecasts from Numerical Weather Prediction models. This shuffling technique consists in reordering a sample such that its rank structure corresponds to the rank structure of a reference sample, and thus allows to reconstruct multivariate dependence structures. The Schaake Shuffle has already been applied for various applications in climate science, such as ensemble postprocessing (e.g., Möller et al., 2013; Schefzik et al., 2013), and in numerous studies (e.g., Voisin et al., 2010; Verkade et al., 2013). According to the "marginal/dependence" category to which it belongs, the $R^2D^2$ method performs first a univariate correction to adjust the marginal distribution of each climate variable. In this study, CDF-t is used for this first step, but it has to be noted that other univariate methods can be employed. Instead of directly applying the Schaake Shuffle and reproducing the temporal structure of the reference (as in Vrac and Friederichs, 2015), the method introduces some variability to the timing of the events, by allowing for the possibility to select a "reference dimension" for the Schaake Shuffle, i.e., one physical variable at one given site, for which rank chronology remains unchanged. Reconstruction of inter-variable and spatial correlations of the reference is then performed using the Schaake Shuffle with the constraint of preserving the rank structure for the "reference dimension". Note that the $R^2D^2$ method can generate as many corrections as the number of variables to be corrected, and all with identical inter-variable and spatial dependencies but with different temporal structures, depending on the selected reference dimension. Hence, $R^2D^2$ introduces some stochasticity in the bias correction. For practical reasons, in the following, we will reduce

the number of corrected outputs: only $R^2D^2$ corrections with reference dimensions located either in Paris or in the center of Brittany (respectively for France and Brittany regions) will be analyzed in Sect. 5. It must also be noted that, by using the Schaake Shuffle technique, $R^2D^2$ assumes by construction the inter-variable and spatial dependence structures (i.e., the rank correlations, or copulas) to be stable in time. Some more mathematical details about $R^2D^2$ are expressed in Appendix B.

### 3.3 "Dynamical Optimal Transport Correction" (dOTC)

The "Dynamical Optimal Transport Correction" (dOTC) method was developed by Robin et al. (2019), in the "all-in-one" category, i.e. correcting the marginal distributions and dependence structures altogether at the same time. Based on optimal transport theory, it is a generalisation of the univariate quantile mapping techniques to the multivariate case. dOTC is aimed at constructing a multivariate transfer function, called a transport plan, to perform bias correction by minimizing a cost function associated with the transformation of a multivariate distribution to another. Multivariate distribution of a biased random variable and its correction are linked together through this particular transfer function, where for any value of the variable to correct is associated a conditional law linking the biased value and its correction. Corrections are then picked (partially) randomly from these conditional laws, introducing some stochasticity into the bias correction procedure.

As for univariate quantile mapping, the way the transfer function is constructed for dOTC plays a decisive role in the obtained bias correction outputs. As explained before, the univariate method CDF-t is able to learn the change of univariate distributions between the calibration and the projection periods for the climate model, and transfers this change to the observational world. Following this philosophy in a multivariate context, dOTC is designed to learn not only the change of univariate distributions, but also the change of multi-dimensional properties of the model, and allows to tranfer them in the corrections. Contrary to $R^2D^2$, it assumes non-stationarity of the dependence (copula) structure between the calibration and the projection periods, and permits to take into account the evolution of the model (e.g., induced by climate change) in the bias correction procedure. More explanations about dOTC are given in Appendix C.

### 3.4 "Multivariate Bias Correction with N-dimensional probability density function transform" (MBCn)

The "Multivariate Bias Correction with N-dimensional probability density function transform" (MBCn) was developed by Cannon (2018) in the context of the "marginal/dependence" category. Based on an adaptation of an image processing algorithm used to transfer colour information, MBCn permits to transfer statistical characteristics of a reference multivariate distribution to the multivariate distribution of climate model variables. Being part of the "marginal/dependence" category, univariate distributions of climate variables are first adjusted using a 1d-BC method. For this step, MBCn uses the Quantile-Delta Mapping method (QDM, Cannon et al., 2015) that preserves absolute or relative changes in quantiles, e.g. for respectively variables like temperature or ratio variables like precipitation. Once univariate distributions are corrected, the dependence structure is adjusted by using an iterative process. At each step, data are multiplied by random orthogonal rotation matrices to partially decorrelate the climate variables to correct. QDM corrections are then applied on (partially) decorrelated data before the recorrelation step with the inverse random matrices. This step (i.e., including rotation, QDM corrections and back rotation) is repeated iteratively until convergence is reached between the multivariate distributions of reference and climate simulations during the calibration

period. Indeed, those iterations permit correcting the dependence structure of the model. Moreover, by doing so – and similarly to dOTC – MBCn allows changes in the dependence structure to be in accordance with the model changes. More details about MBCn can be found in Appendix D.

### 3.5 "Matrix Recorrelation" (MRec)

Bárdossy and Pegram (2012) presented an MBC, hereafter referred to as "Matrix Recorrelation" (MRec). The latter lies in the "all-in-one" category and relies on a matrix recorrelation technique. The MRec method consists in first transforming separately each variable of both model and references to the univariate normal distribution with Gaussian quantile-quantile method. This transformation step is particularly appropriate for variables with mixed distributions (e.g. precipitation composed of wet and dry days), for which computing Pearson correlation matrix on Gaussianized data instead of raw data permit to better describe their

dependence structure. Then, a combination of "decorrelation" and "recorrelation" steps using decompositions of correlation matrices through Singular Value Decomposition (SVD, Beltrami, 1873; Jordan, 1874a, b; Stewart, 1993) is applied on the Gaussianized model data, forcing its Pearson correlation matrix to match that of the Gaussianized observed data during the calibration period. For the projection period, the same "decorrelation-recorrelation" matrix is directly applied on Gaussianized model data, which permits to preserve, for the projection period, the potential changes in correlations as simulated by the model.

Finally, for both periods, a quantile-quantile back transformation is applied separately for each variable between recorrelated variables and references to correct marginal distributions. See Appendix E for more details.

Contrary to the R2D2, dOTC and MBCn methods presented previously, MRec differs in being designed to correct only a particular feature of the multivariate dependence structure, here Pearson correlations. Implicitly, it makes the assumption that Pearson correlation values are sufficient to determine the full multivariate dependence structure, which can be called into

question for variables with skewed and heavy tailed distributions (like precipitation) and with potentially complex interactions that Pearson correlation cannot capture as a whole. For this reason, implementing the MRec algorithm in the present intercomparison study permits to compare the performances of an MBC method based on such an assumption relatively to methods intended to correct the non-Gaussian dependence structure of climate simulations.

## 4  Design of Experiments

### 210  4.1  Settings of MBCs

Multivariate BC methods can be implemented in different dimensional configurations, depending on the need of the users to correct inter-variable and/or spatial correlations. However, in most cases, multivariate BC methods are applied grid cell by grid cell by practitioners to correct inter-variable properties of climate simulations, disregarding spatial structures (e.g., in Meyer et al., 2019; Guo et al., 2019). We tested and assessed this approach for each method, but also expanded the study to include

high-dimensional configurations of MBC to adjust spatial and full (i.e. spatial and inter-variable jointly) dependence structures of climate simulations. Depending on the dimensional configurations, the objectives of corrections for multivariate properties

differ. Including different dimensional versions in the study will permit to better highlight the potential losses and benefits associated with them. Therefore, in the following each of the four MBC methods is applied according to the three following configurations:

– a 2-dimensional (hereinafter referred to as "2d-") version, for which the MBC method is applied independently at each grid cell but jointly corrects both temperature and precipitation time series. For example, to correct a climate dataset of 321 grid cells, the MBC method is performed 321 times, i.e. for each grid cell across the whole grid. By doing so, 2d-versions are aiming to correct inter-variable correlations within each grid cell.

– a Spatial-dimensional (hereinafter referred to as "Spatial-") version, where all time series for a particular physical variable are corrected jointly, but independently from the other physical variable. Hence, for this version, the MBC method is performed twice, adjusting on the one hand all time series for temperature and, on the other hand, all time series for precipitation. Thus, Spatial-versions are designed to adjust spatial correlations of climate models for each physical variable separately.

– a Full-dimensional (hereinafter referred to as "Full-") version, where all time series are corrected jointly over the entire grid for both temperature and precipitation. The MBC method is hence applied only once, and is intended to correct together the inter-variable and spatial correlations of the simulations.

Regarding the initial settings for MBCn, preliminary tests have been conducted with different dimensional settings to find the number of iterations ensuring the converge of the algorithm depending on the dimensional configuration. With respect to the results of these tests (not shown), the number of iterations has been chosen to be equal to 50 for 2d-configurations and 200 for both Spatial- and Full-versions.

### 4.2   Protocols of bias correction

In this study, the BC methods presented above are applied to correct IPSL GCM simulations with either the WFDEI ($0.5° \times 0.5°$) or the SAFRAN (8km×8km) data as references. Data are available for the period 1979-2016, i.e. 38 years, and are divided into two intervals of 19 years: 1979-1997 and 1998-2016. As a reminder, daily temperature and precipitation times are
corrected on 321 and 345 grid cells for France and Brittany regions respectively. For each method, bias correction is performed separately for the 12 months in order to preserve seasonal properties.

The first protocol in this study takes advantage of the cross-validation technique to generate bias corrected outputs for the period 1979-2016. Dividing the time period into two parts permits to perform a two-fold cross-validation procedure: the 1979-1997 period is first defined as the calibration period, and the 1998-2016 portion, called the projection period, is used for
out-of-sample validation. Swapping of the two periods is then done, so that each period has been used once for calibration and once for validation. Bias correction for the period 1979-2016 is then achieved by assembling the adjusted outputs for the projection periods obtained at each step. This 2-fold protocol, largely used in the climate science literature (e.g. in Cannon, 2018), allows to reduce overfitting by using two distinct sub-periods and is hence well suited to evaluate our results. However,

by adjusting the period 1979-1997 according to the 1998-2016 period, this protocol presents the drawback to potentially hide the climate change signal present in the model. Thus, proper assessment of the multi-dimensional properties evolutions cannot be conducted via this procedure.

Hence, to evaluate the non-stationary behavior of BC methods, a second protocol is defined. Similarly to the first protocol, the 1998-2016 period is corrected by using the 1979-1997 portion as calibration period. However, here, 1979-1997 simulations are corrected directly with respect to the 1979-1997 references, i.e., without cross-validation. Hence, the potential climate change signal is not distorted by undesirable effects resulting from the protocol procedure, allowing for the appropriate assessment of change aspects of the BC methods between the two periods.

In accordance with common practice, thresholding of 1 millimetre for precipitation time series is applied before evaluation to replace values lower than 1 millimetre by 0 after correction.

## 5 Results

The correction outputs are evaluated according to different characteristics designed to focus on (i) marginal, (ii) inter-variable, (iii) spatial, (iv) temporal and (v) non-stationary properties. Characteristics (i-iv) are evaluated on the 1979-2016 period for the adjusted outputs obtained according to the 2-fold protocol and are compared to those from the reference dataset. However, regarding non-stationary properties, corrected outputs from the second protocol are used and results are compared to the simulations to highlight the performances of the MBC methods regarding their capability to reproduce (or not) the multi-dimensional changes of the model between the 1979-1997 and 1998-2016 periods.

In the following, evaluation is presented for the winter season (December-January-February) only, as conclusions remain generally the same for the other seasons. However, in order to provide nuances, additional results for the summer season (June-July-August) are displayed in the Supplement when needed.

### 5.1 Univariate distributions properties

First, bias-corrected data are evaluated relatively to univariate statistics. To do so, for temperature and precipitation, the difference of mean values between the bias-correction and the reference at each grid cell is computed. The same computation is also made for standard deviation. Absolute difference is calculated for temperature mean, while relative difference is more appropriate for precipitation mean as well as for standard deviation of both physical variables. Results are shown with boxplots for the plain IPSL simulations and for a selection of BC outputs in Fig. 1 for France during the winter season. The results for Brittany during winter are presented in Fig. S1 of the Supplement. As "marginal/dependence" MBC methods correct univariate properties independently from the dependence structure, results for their 1d-characteristics are equivalent between the three dimensional configurations (2d-, Spatial- and Full-). Therefore, to avoid redundancy, results for $R^2D^2$ and MBCn are presented for only one arbitrary dimensional configuration, the other configurations giving the exact same mean and standard deviation results. Clearly, Fig. 1 shows large differences between the IPSL simulations and the references for both temperature and precipitation, and illustrates the necessity to adjust 1d-distributions of the model before using it in subsequent analyses.

Multivariate BC methods implemented in this study display different performances in adjusting the univariate properties. In agreement with the properties of the "marginal/dependence" MBC methods, $R^2D^2$ and MBCn present exactly the same results as the 1d-BC methods they use, i.e. respectively CDF-t (shown) and QDM (not shown). With regard to the performances of dOTC and MRec, some instabilities are found relatively to the dimensional configuration. For dOTC, increasing the number of dimensions to correct from 2d- to Full- seems to have a slight, but non-negligible, cost on the correction of mean and standard deviation (Fig. 1b and 1c). However, depending on both the climate variable and the statistical feature, the increasing deterioration with respect to the dimensional setting is not systematically observed, as it can be seen in Fig. 1a and 1d. Concerning MRec, a slight deterioration of correction is often observed from 2d- to Spatial- versions (Fig. 1b, 1c and 1d). Regarding the Full-version, the MRec algorithm produces results that are clearly unsatisfactory. Instead of improving the simulations, Full-MRec corrections strongly degrade the univariate statistics. This underperformance of the MRec method over France appears in a context of high-dimensional correction when the number of available data is not large enough compared to the number of dimensions. In this case, the inverses of high-dimensional sample covariance matrices are a highly biased estimator of the inverse of covariance matrices, which consequently largely affects the quality of the Full-MRec corrections. Anyhow, the increasing degradation, whether it is slight or not, of univariate distributions corrections in high-dimensional contexts is one (undesirable) feature of "all-in-one" methods, here observed for dOTC and MRec. Indeed, "all-in-one" methods are designed to adjust both univariate distributions and dependence structure of climate simulations at the same time, which involves a possible deterioration of 1d-marginal distributions during the combined correction process.

For Brittany, the same conclusions hold for $R^2D^2$, dOTC and MBCn, indicating no particular influence of spatial resolution on the results of the marginal statistics adjustment for these methods. Nevertheless, and quite interestingly, for the Full-MRec outputs, the underperformance observed for France is not obtained for Brittany (Fig. S1). A possible reason explaining why Full-MRec version is presenting adequate results on this particular region (and the two other sub-regions, not shown) concerns the size of its geographical area and will be discussed in more details in the subsection 6.2.

## 5.2   Inter-variable correlations

To evaluate inter-variable dependence structure, Spearman correlations between temperature and precipitation are computed at each grid cell to measure the monotonic relationship between the two physical variables. Using rank correlation presents the particularity of not being value-dependent, i.e. it measures the dependence between two variables rid of their univariate distributions. As the goal when applying MBC is to adjust not only the univariate distributions but also the dependence structure between the variables of interest, Spearman's correlation is appropriate for this latter aspect. Moreover, this measure does not require any assumption on the distribution of the variables or their statistical relationships. It is hence appropriate for temperature and precipitation studies presenting extreme values and/or lower bound (Vrac and Friederichs, 2015). The maps of the Spearman correlation differences with respect to the reference – for the IPSL model and the bias-corrected data – are displayed in Fig. 2 for both France and Brittany. Initial maps of Spearman correlations, i.e. without differences with respect to the reference, are also provided in Fig. S2.

For France, map for the IPSL simulations (Fig. 2b1) indicates strong differences with respect to the WFDEI map (Fig. 2a1).

As the univariate CDF-t method does not modify rank sequence of temperature and precipitation time series, it globally con-
serves both the rank correlation intensities and structures of the IPSL model for each region and does not provide any correction
of this aspect (Fig. 2c1). By construction, clear improvements of the inter-variable correlation structure are provided by 2d-
versions (Fig. 2d1, 2g1, 2j1 and 2m1). This is also the case for most of the Full-configurations of MBCs (respectively, Fig. 2f1,
2i1, 2l1) despite possible differences in intensities. Note that maps of correlation differences for 2d-$R^2D^2$ (Fig. 2d1) and Full-

$R^2D^2$ (Fig. 2f1) are identical. Indeed, for the inter-variable aspect, 2d-version is nested within the Full-configuration (see Vrac,
2018), due to the use of the reordering technique in $R^2D^2$. Also, for $R^2D^2$, the choice of the reference dimension does not have
any impact on results in the inter-variable context, as it only modifies the rank chronology of time series. As expected from pre-
vious explanations, the map for the Full-version of MRec (2o1) indicates a strong deterioration of the inter-variable correlation
structure. It highlights again the inability of the method to work properly for France in this dimensional setting. Concerning

Spatial-versions of MBCs (Fig. 2e1, 2h1, 2k1 and 2n1), as they adjust the whole simulated field of temperature and precip-
itation separately, they disregard inter-variable relationships. It results in BC outputs with strongly weakened inter-variable
correlations structures.

Regarding Brittany, the same conclusions can be drawn for $R^2D^2$ and dOTC, for which spatial resolution does not affect
the results of inter-variable properties adjustment. As noted previously, Full-MRec over Brittany provides more satisfactory

results than those obtained over France, and are in line with those obtained for $R^2D^2$ and dOTC. However, for MBCn outputs,
a degrading effect from 2d- (Fig. 2j2) to Full- (Fig. 2l2) is observed, in providing a corrected correlations' structure but with
underestimated intensities in the high-dimensional context.

## 5.3   Spatial correlations

To assess the quality of the corrections in terms of spatial correlations, mean correlograms, i.e. mean Spearman correlation

in function of distance, are computed for temperature and precipitation separately after removing daily areal mean. Indeed,
climate variables can present a high day-to-day variability that can affect the evaluation of spatial criteria if not removed (e.g.,
Vrac, 2018).

Figure 3 and S3 show the results obtained for respectively precipitation and temperature for the different climate datasets.
Note that the choice of the reference dimension for $R^2D^2$-versions modifies results for temporal criteria, and consequently

for some of the spatial criteria. Hence, in the rest of this work, results from $R^2D^2$-versions are presented with the reference
dimension corresponding to the variable under interest. For sake of brevity, results for precipitation are mainly discussed in
this subsection and nuances are made when different results are obtained for temperature.

For France, IPSL precipitation correlogram is fairly distinct from WFDEI one. The univariate method CDF-t, by simply
adjusting univariate distributions, gets closer to the reference dataset (Fig. 3a1), which may be here confusing. Indeed, although

CDF-t adjusts the univariate distributions, it is supposed to preserve the rank sequence of the simulations and therefore spatial
correlations are disregarded during the BC procedure. But, as the Singularity Stochastic Removal version of CDF-t (Vrac et al.,
2016) is explicitly designed to improve dry days frequency, the method consequently modifies rank correlations, which results

here in an improvement of spatial statistics for precipitation. Also, an additional reason is that the correction of the univariate distributions provided by CDF-t associated with the removing of daily areal means modifies ranks of the data, resulting in getting a correlogram closer to that from the reference dataset, and so improves inter-site variability.

Correlograms of 2d-versions (dotted) for the four MBC methods (Fig. 3b1, 3c1, 3d1 and 3e1) show results equivalent to CDF-t. Indeed, 2d-configurations MBCs adjust univariate distributions and inter-variable correlations without modifying spatial correlations. The improvements of correlograms for 2d-versions thereby illustrate again that the correction of univariate distributions improves spatial statistics for France. Particularly, 2d-$R^2D^2$ results (Fig. 3b1) are, by construction, exactly the same as those from CDF-t (Vrac, 2018). Indeed, by construction, 2d-$R^2D^2$ driven by precipitation preserves Spearman spatial correlations from CDF-t for the precipitation variable. Note that, however, it is definitely not the case for temperature spatial structure (not shown) when 2d-$R^2D^2$ is driven by precipitation. Indeed, for 2d-$R^2D^2$ outputs driven by a specific physical variable, spatial structures of the "other" variable are strongly degraded by the reordering step.

Correlograms associated with Spatial- and Full-versions outputs for $R^2D^2$ (Fig. 3b1) nicely fit the one from the reference dataset – even at long distances – and provide major improvements in adjusting the spatial properties of the simulations. However, for similar reasons as those explained for 2d-$R^2D^2$, undesirable degradation effects on spatial cross-correlation between temperature and precipitation are obtained for Spatial-$R^2D^2$ outputs (not shown). Therefore, it indicates that practitioners must favour the use of Full-$R^2D^2$ for their applications. With regard to Spatial- and Full-dOTC (Fig. 3c1) and Spatial-MRec (Fig. 3e1), although correlograms are very close to those from the reference dataset, they provide slightly less pronounced improvements compared to the 2d-versions, suggesting a slight degrading effect on results for these methods by considering more variables in the correction. As expected, the correlogram associated with Full-MRec outputs is away from reference data, indicating once again the dysfunction of the MRec method for France. For Spatial- and Full-MBCn (Fig. 3d1), at long distances, similar improvement of spatial correlations are provided as those from dOTC. However, large deviations between correlograms are found for short distances, suggesting a failure for the MBCn method to adjust local spatial properties in a high-dimensional context.

For Brittany, same conclusions hold for $R^2D^2$ (Fig. 3b2), presenting again a stability of results regardless of both the spatial resolution and the geographical area considered. For dOTC (Fig. 3c2), Spatial- and Full-versions now provide major improvements of spatial correlations compared to their 2d-versions and present results similar to Spatial- and Full-$R^2D^2$. With regard to MRec (Fig. 3e2), the dysfunction of the Full-version is no longer observed. It now provides results similar to Spatial-MRec and better than 2d-MRec. However, it is worth mentioning that, for Brittany, different results are obtained with MRec between precipitation and temperature spatial corrections. While, for temperature, Spatial-MRec outputs (Fig. S3e2) provide reasonable results with a correlogram relatively close to the one of the reference data, a more moderate improvement of inter-site variability is obtained for precipitation (Fig. 3e2). Explanations for these results will be provided in subsection 6.2. Regarding MBCn (Fig. 3d2), large deviations between correlograms are found for both short and large distances, underlining some instability of the algorithm to adjust for spatial correlations.

## 5.4 Temporal structure

The different MBC methods implemented here, are not intended to adjust temporal structures. Indeed, these multivariate procedures adjust multivariate distributions without accounting for any temporal information. However, although the temporal structures are not adjusted according to the reference, MBCs necessarily modify the rank sequences of the simulations (Vrac, 2018). This modification is not performed in the same way depending on the MBC or the dimensional configuration used, and remains therefore to evaluate. To do so, 1-day lag Pearson autocorrelations are computed at each grid cell for temperature and precipitation. The resulting maps of differences with respect to the reference for the different climate datasets are displayed in Fig. 4 (resp. Fig. S4) for temperature (resp. precipitation).

For France, IPSL temperature autocorrelations differences (Fig. 4b1) are small, indicating a relative agreement of IPSL with WFDEI reference dataset (Fig. 4a1), showing equivalent high values. Similar differences map are provided by CDF-t outputs (Fig. 4c1). It is however not the case for precipitation (Fig. S4c1), for which a decrease of autocorrelation values is observed over France with respect to the reference and to the model. Although not observed for temperature, it highlights that the univariate correction could have a non-negligible effect on Pearson autocorrelation. Interestingly, 2d-versions (Figs. 4d1, 4g1, 4j1 and 4m1) do not lead to a strong modification of temporal properties with respect to CDF-t. However, from one method to another, temporal structure modifications are not equivalent for Spatial- and Full-versions. For dOTC and MBCn (Figs. 4h1, 4i1, 4k1 and 4l1), as the number of dimensions increases, the temperature autocorrelations seem to be more and more modified, with intensities of values decreasing slightly from Spatial- to Full-versions. This result can also be seen for precipitation in Fig. S4. With regard to MRec, its Spatial-version (Fig. 4n1) presents similar results than those obtained from Spatial-dOTC and Spatial-MBCn. Also, and as expected, Full-MRec outputs (Fig. 4o1) do not provide sensible results due to the inability of the method to work properly over the whole France. Concerning $R^2D^2$, as the reference dimension driving the rank sequence is the same between Spatial- and Full-configurations, same differences of autocorrelation maps are obtained for these two versions (Figs. 4e1 and 4f1). Moreover, autocorrelation value in the grid cell of the reference dimension, i.e. located over Paris for France, is exactly equal to the corresponding one in the CDF-t outputs, by construction. Remarkably, as mentioned by Vrac (2018), autocorrelations of the CDF-t outputs are partially reproduced around the specific locations of the reference dimensions for Spatial-$R^2D^2$ and Full-$R^2D^2$ versions, as evidenced by the lightly-shaded area around Paris. This reflects the existing spatial correlations between the reference dimension and its local neighbourhood, which results in partially reproducing the temporal properties of the model over this area. However, for precipitation (Figs. S4e1 and S4f1), this result is not as clear-cut as it is for temperature, probably due to weaker spatial correlations around Paris for this physical variable.

In a general way, the same conclusions can be drawn for Brittany, sometimes even better illustrated due to a narrower color scale. The results for Full-MRec are easier to interpret. They present results similar to those from 2d- and Spatial-MRec (Fig. 4o2). In particular, it indicates that, contrary to dOTC and MBCn, MRec does not present an increasing modification of temperature autocorrelations from 2d- to Full-versions.

To better understand the results obtained from Fig. 4, further explanations are required. The relative agreement of Pearson autocorrelation values between the reference and IPSL dataset showed by Fig. 4 might lead one to believe that temporal

properties of the model are quite correct for temperature, which is in reality misleading for two main reasons. First, 1-day lag Pearson autocorrelation permits to assess only a particular feature of the temporal properties, which is obviously insufficient to draw any general conclusions on the quality of simulations concerning these aspects. For example, by simply computing Pearson temperature autocorrelations for higher lag values, a discrepancy of results is obtained between the reference and the simulations (not shown). Second, Pearson autocorrelations depend on two statistical characteristics of time series: their variability and their temporal rank structures. As implemented in Fig. 4, the Pearson autocorrelation metric is hence not able to dissociate them. The similarity between reference and model autocorrelations can then potentially be the combined result of errors stemming from both biased univariate distributions and wrong rank structures of the model.

To better assess temporal structure changes brought by MBCs, the calculation of rank correlations between the bias corrected time series and the raw climate model simulations is performed for each physical variable and at each grid cell. Results for temperature and precipitation are displayed with boxplots, respectively in Figs. 5 and S5. The closer the values of the boxplots are to 1, the closer the rank chronologies of the MBC outputs are to the rank chronologies of the model. For France, as expected, similar temperature rank structures are observed between the model and CDF-t/2d-$R^2D^2$ outputs (Fig. 5a). For the other 2d-versions, rank correlation values are quite close to 1 as well, suggesting that dOTC, MBCn and MRec methods in their 2d-configuration modify only slightly the rank structure of the initial simulations. For Spatial- and Full-configurations, dOTC and MBCn change moderately the rank structures even though they consider more dimensions in the correction. Concerning MRec, without analysing the Full-outputs, the increasing modification with dimensionality is also observed between 2d- and Spatial-MRec outputs, although less pronounced. In contrast, for Spatial- and Full-$R^2D^2$ outputs, the changes in the rank structures for France are substantially larger than those discussed until now. This result is also obtained for precipitation in Fig. S5a with an even larger range. The principal reason lies in the fact that, as already explained, $R^2D^2$ partially preserves rank sequences of the CDF-t outputs – and therefore of the IPSL model – in the direct neighborhood of the reference dimensions, but strongly modifies the rank structures outside this neighbourhood, which results in obtaining some low Spearman correlation values in Fig. 5a and Fig. S5a .

For Brittany, results show a less pronounced modification of rank structure for both temperature (Fig. 5b) and precipitation (Fig. S5b) than those observed for France. In particular for temperature, similar rank correlations are obtained for all versions of the methods, even for Spatial- and Full-$R^2D^2$ outputs, indicating that the number of dimensions has potentially a non-significant effect on this criteria over a smaller area. The differences of results between France and Brittany highlight that the size of the region of interest seems to have a non-negligible influence on the temporal properties of BC outputs.

## 5.5 Multi-dimensional changes analysis

When correcting climate simulations, in practice, while climate simulations for the present period are adjusted with respect to observations, no reference data are available for the correction of future periods. Assumptions of either stationarity or non-stationarity of copula are then made within the MBCs concerning the change of the multi-dimensional features between present and future periods. This has then consequences on how MBCs can account for the changes in the multidimensional properties of the climate simulations. Therefore, using the second protocol defined in subsection 4.2, we now focus on how the different

MBC methods reproduce the change of inter-variable and inter-site structures, as given by the model to be corrected between two different periods.

### 5.5.1 Analysis of change in inter-variable correlations

Fig. 6 shows, for the bias-corrected outputs, the maps of the difference between the Spearman correlation between temperature and precipitation, computed for the calibration (1979-1997) and the projection (1998-2016) period, respectively. It permits to visually assess part of the change in the inter-variable dependence structure. Over France, inter-variable change of the IPSL simulations (Fig. 6b1) seems to be distinct from those of WFDEI (Figs. 6a1). CDF-t outputs (Fig. 6c1) reproduce globally the change of the simulations, as they present similar maps. Concerning results for the 2d-(Fig. 6d1) and Full-versions (Fig. 6f1) of $R^2D^2$, they present inter-variable rank correlation values close to 0. This illustrates the stationarity assumption in $R^2D^2$: the copula function (i.e., dependence structure) of the observations during the calibration period is reproduced for both calibration and projection, resulting in having no change of inter-variable rank correlations. For their part, 2d-dOTC, 2d-MBCn and 2d-MRec maps (resp. Figs. 6g1, 6j1 and 6m1) present roughly the same spatial structures for the differences of Spearman correlations, which indicates that the evolution of the simulations is somehow taken into account in the correction procedures. It must be remarked that, contrary to dOTC and MRec, the stochastic generation of random rotation matrices within the MBCn algorithm leads to get a non-negligible variability in the estimation of the evolution (not shown). This highlights a particular aspect of MBCn: contrary to other methods, MBCn is based on a stochastic procedure, which has a significant impact on its adjustments. Consequently, the quality of MBC data obtained from MBCn can differ from a correction to another for the same climate simulation, depending on the random rotation matrices generated in the algorithm and on the stopping rule (i.e., number of iterations). Interestingly, concerning the methods Spatial-versions (Figs. 6e1, 6h1, 6k1 and 6n1), outputs show changes of inter-variable rank structure similar to those from the model. Indeed, and as for CDF-t, rank inter-variable correlations are not adjusted with Spatial-versions. Consequently, the change of inter-variable rank structure of the model is somehow preserved in Spatial-versions outputs.

For the Full-configuration maps of dOTC and MBCn (Figs. 6i1 and 6l1), changes simulated by the model are not reproduced at all, which might be due to the failure of these methods to handle the change in time of this statistical feature in high-dimension. As expected, Full-MRec map (Fig. 6o1) does not provide adequate results due to its inability to adjust the simulated data for France in this dimensional setting.

Concerning the results for Brittany, conclusions similar to those obtained for France can be drawn for $R^2D^2$ outputs. However, conclusions are quite different for CDF-t, 2d-dOTC, 2d-MBCn and 2d-MRec. Indeed, the changes of rank correlations obtained for these outputs (Figs. 6c2, 6g2, 6j2 and 6m2) are not in agreement at all with the simulated ones (Fig. 6b2). In fact, changes from 2d-outputs are in line with those from CDF-t, illustrating the importance of the correction of 1d-characteristics for inter-variable changes. It is also the case for the Full-MRec map (Fig. 6o2), providing more sensible results than those obtained for France.

Generally speaking, for 2d- and Spatial-versions of MBCs making the assumptions of copula non-stationarity, similar results than those brought by their univariate BC outputs are obtained, suggesting the importance of the correction of univariate

distributions for changes of inter-variable rank correlations. Additional results in agreement with these conclusions are obtained for summer and are displayed in Fig. S6.

## 5.5.2 Analysis of change in spatial correlations

In order to assess changes of spatial structures in bias-corrected outputs, $p$-Wasserstein distance (see, e.g., Villani, 2008, chap. 6) is computed. This metric measures the distance between two multivariate probability distributions $\mu$ and $\upsilon$, and is defined as follows:

$$W_p(\mu, \upsilon) := \left( \inf_{\gamma \in \tau(\mu, \upsilon)} \int_{\mathbb{R}^d \times \mathbb{R}^d} ||x - y||^p \mathrm{d}\gamma(x, y) \right)^{\frac{1}{p}}, \tag{1}$$

with $\tau(\mu, \upsilon)$ denoting the set of probability measures on $\mathbb{R}^d \times \mathbb{R}^d$ with respectively $\mu$ and $\upsilon$ as first and second margins, and $||.||$ the Euclidean distance. In the present study, $p$ is taken equal to 2, as it ensures the uniqueness of the minimization problem (Santambrogio, 2015). The Wasserstein distance can be seen as the minimum "cost" for transforming a multivariate probability distribution $\mu$ into another, here $\upsilon$. In particular, computing Eq. (1) between a distribution characterising a sample during the calibration period and another distribution characterising a sample during the projection period, permits to provide information on its change across time, whether it represents a univariate, multi-variable or multi-site (or both) distribution. More details on how to compute in practice this distance are provided in Appendix C. The resulting metric, denoted Wd, is calculated using the R package 'transport' (Schuhmacher et al., 2019) over the region of interest according to three different multivariate distributions:

- on ranks of temperature only over the whole region to assess change in the spatial dependence structure of temperature;

- on ranks for precipitation only over the whole region to assess change in the spatial dependence structure of precipitation;

- on ranks for both temperature and precipitation over the whole region to assess change in the inter-variable and spatial dependence structures of the two variables.

In particular, computing Wd using ranks instead of raw values allows removing the change in the univariate distributions from that in spatial and inter-variable relationships. However, comparing Wd values of climate datasets must be made with caution. Indeed, similar values of Wd for different climate datasets do not necessarily imply that their changes of spatial structure are similar. Results for the three Wasserstein distances on ranks are displayed in Fig. 7 for both France and Brittany. Additional results for Wd on raw values are displayed in Fig. S7 for information purposes only.

For France (Fig. 7a), the three Wd are slightly higher for the reference than for the model data (represented by straight lines). Although the differences are quite small, it cannot be concluded directly that changes of spatial structure are identical, as there is no particular reason for this. For CDF-t outputs, similar Wd are obtained as those from the model. However, as the 1d-BC method does not modify (too much) rank sequence of temperature and precipitation time series, it can be deduced that CDF-t outputs globally reproduce/preserve the spatial structure change of the model.

For 2d-$R^2D^2$ outputs, two results are presented, corresponding to those obtained with either temperature or precipitation used as reference dimension. For the reasons already given (see e.g. subsection 5.3), results for 2d-$R^2D^2$ driven by temperature

(resp. precipitation) for the change of spatial structure of temperature (resp. precipitation) are by construction identical to those from CDF-t. Nevertheless, for the spatial structure of temperature and precipitation jointly (triangle), Wd for 2d-$R^2D^2$ outputs are quite high. Indeed, when 2d-$R^2D^2$ version uses either temperature or precipitation rank sequence to drive the "other" physical variable at each grid cell, the method is likely to degrade the spatial structures of the "other" variable in a different way for calibration and projection periods. Consequently, the Wasserstein distance captures a "change" in the spatial structure

of the two variables between these two periods, but is in fact due to its deterioration. Concerning Spatial-$R^2D^2$, low Wd are observed for the change of the spatial structures for temperature and precipitation separately, illustrating the stationarity copula assumption used. However, for the Wd computed for the whole multivariate distribution (triangle in Fig. 7a), Spatial-$R^2D^2$ presents a higher value, close to that of the IPSL simulations. Indeed, as already explained in 5.5.1, within Spatial-$R^2D^2$, copula functions of temperature and precipitation are adjusted separately without correcting inter-variable rank correlations,

which results in partially preserving the changes of inter-variable rank structure of the model between calibration and projection period. With regard to Full-$R^2D^2$, the three Wd are all quite low, in agreement with the stationarity copula assumption it uses. However, it should be noted that the Wd are not equal to 0, whereas, theoretically, no change of spatial structure is performed by Full-$R^2D^2$. In addition to the reason already cited concerning dry days frequency correction, this is also due to the fact that, in the present study, bias corrections have been performed on a monthly basis, while the evaluation is done at a seasonal scale.

For both dOTC and MBCn outputs, Wd are higher than those from the model. Although the changes in spatial correlations derived by these two methods are too strong, it nevertheless highlights their ability to capture such a change from the model and to use it in their bias correction procedure. Moreover, as explained in subsection 5.4, dOTC and MBCn methods modify only slightly the rank structure of the initial simulations. It can then be deduced that the changes in spatial correlations measured for the two methods are (partially) in agreement with those from the model. However, for MBCn, the three Wasserstein distances

increase according to the number of dimensions considered in the bias correction, from 2d- to Full-versions. It can be linked with the deterioration of the quality of results already observed for spatial features for very high-dimensional bias correction. Regarding MRec, and without speaking about its Full-version, similar observations can be made for 2d- and Spatial-outputs as well. In a general way, the Wd associated to the different configurations for dOTC, MBCn and MRec are always above the Wasserstein distances for $R^2D^2$, illustrating somehow the assumptions made by these methods about the stationary or

non-stationary copula functions.

For Brittany (Fig. 7b), the Wd values computed for the model are quite low, indicating little simulated change of spatial structures for this region. Consequently, the differences of Wd between methods assuming stationarity and non-stationarity of copula functions are less pronounced, but the same conclusions as those drawn for France hold. However, for Full-MRec outputs, Wd values are in relative agreement with those from the model, highlighting the ability of the method to preserve

(partially) the simulated changes of spatial structure between the calibration and the projection periods, for a smaller region.

# 6    Conclusion, discussion and future work

## 6.1    Conclusion

In this study, we have presented a global picture of the performances of four multivariate bias correction (MBC) methods designed to adjust various multivariate properties of climate simulations. These MBC methods were carefully selected for their differences in terms of methodologies, statistical techniques used, assumptions and philosophical features. For each method, three dimensional configurations have been tested to correct climate simulations from the IPSL model: a "2d" version to adjust temperature and precipitation time series together but separately for each grid cell, a "Spatial" version aiming to correct the simulated fields of temperature and precipitation separately, and a "Full" version designed to adjust the two physical variables jointly over the entire domain. Depending on the versions, the objectives of adjustments for multivariate properties are not the same: whereas "2d" and "Spatial" versions are designed to correct respectively inter-variable and inter-site dependence structures, it is expected from the "Full" versions to adjust both the inter-variable and inter-site relationships together. In addition, the univariate CDF-t bias correction method has been implemented and used as a benchmark to assess the benefits of considering multivariate aspects in the correction procedure. A wide range of metrics has been developed to compare bias correction outputs with observations and model data, and analyze the adjustements of univariate distributions, inter-variable correlations, inter-site correlations and temporal structure. Multi-dimensional change, i.e. non-stationary, properties have been assessed, providing a comprehensive framework to compare the performance of the methods. The IPSL simulations have been corrected with respect to two distinct reference datasets, i.e. WFDEI and SAFRAN, for respectively France and Brittany to attempt to measure the potential influence of the reference spatial resolution on MBC results.

## 6.2    Discussion and recommendations

General recommendations can be drawn to help practitioners in the choice of BC methods for their applications. For sake of clarity, Table 2 provides a concise summary of the different recommendations made below. If the univariate CDF-t method corrects the univariate distributions well, it replicates the dependence properties of the model, i.e. inter-variable, inter-site or temporal structures, and preserves its multi-dimensional change across time. Hence, if the multivariate properties of raw climate simulations are not relevant, using 1d-BC methods is not appropriate to get adequate dependence properties. Concerning MBC methods, in general, $R^2D^2$, dOTC, MBCn and MRec algorithms showed a great ability to adjust the statistical properties associated with the corresponding objectives of the dimensional configurations. Indeed, in addition to correcting univariate distributions, the 2d-, Spatial- and Full-versions of each multivariate method adjust respectively inter-variable, spatial, and inter-variable/spatial correlations of climate simulations reasonably well. However, caution has to be taken before applying multivariate methods and conducting analysis studies. It has been noted that, depending on the dimensional configuration, instability of some methods can possibly affect corrected outputs, and practitioners have to make sure that no degradation of the desired statistical features is made by the multivariate BC method. In particular, for MBCn and MRec, increasing the number of variables to be corrected jointly in the dimensional configuration is often accompanied by a potentially strong deterioration of spatial properties (see orange tildes in the row "Capacity to correct spatial prop." in Table 2). However, for

MBCn, it must be recalled that the number of iterations for the algorithm was fixed to 200 for Full-versions. Although this choice is a good compromise between computation time and fitting the multivariate distribution in the calibration period, this might be sub-optimal for some regions. Indeed, early stopping of the procedure could be necessary to avoid overfitting in high dimension, as discussed in Cannon (2018). Therefore, more research is needed to improve the global performances of MBCn, such as early stopping, optimizing the sequence of random rotation matrices to speed up convergence, or, for spatial downscaling problems, adding a conservation step to provide more physical constraints to the bias correction (as proposed in Lange, 2019). Moreover, it has been shown that the characteristics of the climate data to correct can influence the results of the MBCs. In particular, and as noted in subsection 5.3, a distinction of results between temperature and precipitation has been identified for the MRec method (e.g. in Figs. 1, S1, 3 and S3). This might be caused by the way the MRec method performs the correction: only the Pearson correlation structure is adjusted, since assumed to be sufficient to correct the full multivariate dependence structure. Although correcting only Pearson spatial correlations for temperature seems reasonable as temperature has traditionally a multivariate Gaussian dependence structure, it appears to be not enough for precipitation, presenting more complex spatial interactions. In that sense, to adjust non-Gaussian climate variables as precipitation, MBCs correcting the full multivariate dependence structure (e.g. $R^2D^2$, dOTC or MBCn) must be preferred by practitioners.

Also, the ability of the MRec method to adjust Brittany in a very high-dimensional context strongly suggests that the size of the geographical area under study is an important feature for multivariate bias correction. Indeed, a small region like Brittany is likely to present a homogeneous climate, or at least to be spatially second order stationary, and, consequently, strong statistical dependencies between locations. Dimensions are then somehow redundant and spatial correlations for each physical variable are strong, which potentially reduces the number of "effective dimensions", also called "spatial degrees of freedom" (e.g., in der Megreditchian, 1990; Bretherton et al., 1999). For MRec, it results in consequently reducing the errors in the computation of the inverse covariance matrices, and providing more adequate results. For larger regions presenting a high number of "effective dimensions" as France, MRec is however able to provide appropriate results if enough data are provided. For illustration purposes, the MRec method has been additionally applied on a seasonal basis instead of on a monthly one, i.e. correcting 642 dimensions with at least 90 days $\times$ 19 years = 1710 time steps. By increasing the number of time steps used in the procedure, high-dimensional sample covariance matrices within MRec are estimated in a more "robust" way, permitting a more suitable correction of the simulations using Full-MRec. Results for some criteria are presented in the Supplement (Figs. S8, S9, S10, S11 and S12) but are not commented in the present study. Also, within MRec, more robust estimators of inverse covariance matrices could be used to obtain more appropriate corrections in a high-dimensional context (e.g., as presented in Levina et al., 2008). More generally, for most MBCs, for a given number of statistical dimensions (e.g., number of grid cells), as going from a large (e.g., France) to a smaller (e.g., Brittany) area reduces the "effective dimension", it facilitates the multivariate corrections and therefore improves the results (e.g. compare Figs. 1, S1, 4, S4, 5 and S5). This raises the question of whether applying MBC on climate simulations over large geographical areas is justified, i.e. if it is worth striving for the correction of correlation structures between distant sites presenting weak statistical relationships, and, by doing so, taking the risk of losing global effectiveness of the BC methods. It also highlights the importance of choosing parsimoniously the variables to correct,

in order to adjust dependence structures that are relevant without potential quality loss induced by additional (and unneeded) variables.

Regarding the temporal structure, none of the presented multivariate BC methods are designed to adjust this specific statistical aspect (red crosses in Table 2). Moreover, as highlighted by Vrac (2018), any multivariate BC method will necessarily modify the rank sequence of the simulated variables. Results from the present study allow adding nuances to this statement: modification of rank chronologies of the simulations depends on both the multivariate BC methods and the dimensional configurations. In particular, for dOTC, MBCn and MRec methods, a similar behaviour was observed: the higher the number

of dimensions to correct, the stronger the deterioration of rank chronology of the simulations. However, concerning $R^2D^2$, depending on the dimensional version, the rank chronology of the model can be reproduced for the specific area around the location of the reference dimension, which could (or not) be desired by practitioners depending on the performance of the simulations.

      Finally, we shed light on the non-stationary properties of the multivariate BC methods. While dOTC, MBCn and MRec

are designed to transfer some of the multi-dimensional properties evolution (i.e., change in time) from the model to the bias-corrected data, $R^2D^2$ assumes the inter-variable and inter-site rank correlations – or copula functions – to be stable in time. In a general way, copula non-stationarity for future periods can be reasonably expected, e.g. as documented for rainfall spatial distributions (Wasko et al., 2016), for the dependence between storm surge and rainfall (Wahl et al., 2015) and the dependence between seasonal summer temperature and precipitation (Zscheischler and Seneviratne, 2017). However, on the contrary, it

can be argued that inter-variable and spatial dependence structures can be assumed to be stable over time for specific regions, because, to some extent, they can be considered as imposed by physical regional constraints (Vrac, 2018). The differences of Wasserstein distances between the France and the Brittany region for the reference in Fig. 7a and 7b illustrate well that copula stationarity (or non-stationarity) is not straightforward depending on the geographical domain. The question of the evolution of the copula (i.e., the rank dependence structure) is, therefore, still an open question and needs to be answered on a case by

case basis. In practice, performances of the methods concerning the multi-dimensional changes of the different BC outputs are hard to assess precisely, as the potential instability (as in MBCn and MRec) or the stochasticity (as in MBCn) of the methods could affect the quality of the results, making difficult the identification of changes. Moreover, the adjustment of univariate distributions has a non-negligible effect on changes of inter-variable and spatial rank dependences for MBCs assuming non-copula stationarity: in fact, rather than reproducing simulated changes in the correction procedure, these methods are more

likely to provide changes in agreement with the ones provided by 1d-BC (e.g. as seen for Brittany in Fig. 6b). Then, in the case where the adjustement of univariate distributions does not modify (too much) the simulated changes of inter-variable and spatial rank dependences, MBCs assuming non-stationary copula would be more likely to present changes in line with those from the model. This result is further confirmed by the results obtained for summer and displayed in Fig. S6 for inter-variable rank dependence changes. The non-stationary property also partly explains the possible differences of results obtained during

evaluation (i.e., protocol 1, see Sect. 5) for each criterion. Indeed, as noted in Robin et al. (2019), if the multivariate properties changes provided by the model simulations are incorrect, those of the corrections from methods assuming non-stationarity can be, retrospectively, in disagreement with the changes of the observations.

Therefore, before choosing any multivariate BC method, practitioners have to ask themselves some questions, e.g.: What are the important statistical properties I want my corrections to provide? Can the evolution of the copula (i.e., rank dependence) in the simulations between calibration and projection be considered as relevant? And should it be reproduced in the correction? If so, according to the results obtained in the present study, dOTC and MRec are good candidates among the presented MBCs. Using these methods, the corrections will be likely to present change in rank dependence similar to the simulations, or at least of same sign. It could also be recommended to use these methods if practitioners do not have any idea if the rank dependence changes in the simulations could be considered relevant or not, advocating to let the model express its own dynamic in the absence of relevant judgements. However, if it is assumed that the change in the simulations, in spite of all efforts exerted by climate modellers, is not considered as relevant, $R^2D^2$ is a good candidate, as it is better to have stationarity of multi-dimensional rank properties in the correction rather than a non-relevant or wrong one. Moreover, $R^2D^2$ is also a good candidate for practitioners who do not expect any rank dependence change. The obtained BC outputs from $R^2D^2$ will not have any change of inter-variable or inter-site rank dependence structures, because assumed to be imposed by physical constraints, and hence stable in time. Concerning MBCn, the global instability of the method in high-dimensional settings, added to the inherent variability due to its stochastic nature affect significantly the quality of the correction. In practice, therefore, it makes difficult the appropriate preservation of the simulated changes, although the method is specifically designed for that.

## 6.3 Future work

This intercomparison has been designed such that new BC methods can be easily added. As a result, adding new methods relying on different assumptions, correcting different statistical aspects or using other statistical techniques, is reasonably feasible. Moreover, as mentioned in the introduction section, bias adjusted simulations are particularly valuable for impact studies. Despite the challenge of missing impact data, evaluating how the quality of multivariate bias-corrected data influences the results of complex impact models is an important perspective. Providing such an analysis will be useful for the scientific community working on climate change impacts, e.g., in hydrology, agronomy or ecology. In an attempt to answer this question, an appropriate future step could be to apply the presented multivariate BC methods in different dimensional configurations to various GCM simulations – and not only one as in this study – in order to provide an ensemble of multivariate BC simulations. The obtained datasets would also be useful to carry out scientific studies on other aspects of climate change, such as climate change attribution studies aimed to identify which mechanisms are responsible for changes in the Earth's climate (e.g., Stott et al., 2016; Yiou et al., 2017; Ribes et al., 2019). Indeed, most of these studies use plain simulations, and consequently do not take into account their statistical biases. Conducting attribution studies using plain and bias-corrected simulations will permit to increase the understanding of the influence of these biases on results, which is essential to provide valuable information to the society concerning the ongoing climate change.

In the present study, it has been highlighted that none of the presented multivariate BC methods were designed to correct or preserve the temporal properties of the simulations. Nevertheless, a few studies have attempted to develop BC methods providing adjustments of some temporal properties of climate variables in addition to the correction of inter-site or inter-variable properties (Mehrotra and Sharma, 2015, 2016, 2019). However, considering adjustments for temporal properties will

necessarily modify, even slightly, univariate distributions, inter-site and/or inter-variable properties. From a more philosophical perspective, striving for the development of MBCs correcting a wide range of statistical features raises also the question of what has been preserved from the simulations in the final BC outputs. By improving the agreement of simulations with observations, this may have the effect of lowering (misleadingly) the uncertainty of the simulated statistical attributes, often without sound physical justifications (Ehret et al., 2012), which puts into question the validity of such methods. Multivariate BC methods developed in the future should, therefore, take into account these issues, in attempting to find a reasonable balance between, on the one hand, the correction of inter-site and inter-variable dependences and, on the other hand, the correction or modification of temporal properties, while being able to preserve meaningful simulated characteristics for future periods. To do so, developing new MBC methods including some physical processes to drive the correction procedure is a consistent perspective of development to obtain more realistic bias corrected simulations. The new developed MBCs could be then included in this intercomparison study, to evaluate and compare their performances with the existing multivariate BC methods.

## Appendix A: Details on the CDF-t method

BC methods are applied to correct a simulated fields of $S$ grid cells, each of them described by $V$ physical variables. The total number of statistical dimensions to correct is hence equal to $D = V \times S$, with each of the dimension composed of $N$ time steps. Let $\mathbf{X}_A$ being a matrix of dimension $N \times D$ and $X_A^d(t)$ the value of the physical variable corresponding to the $d$-th dimension at time $t$ from the matrix $\mathbf{X}_A$. Datasets, i.e. matrices, to correct with BC methods are model outputs during the calibration (denoted $\mathbf{X}_{M_C}$) and the projection period (denoted $\mathbf{X}_{M_P}$), according to the data from the reference observed during calibration (denoted $\mathbf{X}_{R_C}$). Corrected outputs for the calibration and the projection period are denoted $\widehat{\mathbf{X}}_{M_C}$ and $\widehat{\mathbf{X}}_{M_P}$ respectively.

CDF-t is a version of quantile-quantile method that takes into account, by defining a transfer function $T$, the potential evolution of univariate CDFs from the calibration to the projection period. Let assume for this subsection $F_{M_C}^d$ and $F_{R_C}^d$ defining the univariate CDFs of the $d$-th dimension $X_{M_C}^d$ and $X_{R_C}^d$ located at the same grid cell for both the model and the reference in the calibration period. To simplify the notation, we will denote these CDFs $F_{M_C}$ and $F_{R_C}$ respectively. The transfer function $T$ is defined such that it links the two CDFs $F_{M_C}$ and $F_{R_C}$ as following:

$$T\big(F_{M_C}(x)\big) = F_{R_C}(x). \tag{A1}$$

A more simple formulation of $T$ is then obtained by replacing $x$ by $F_{M_C}^{-1}(u)$, with $u$ probabilities in $[0,1]$.

$$T(u) = F_{R_C}\big(F_{M_C}^{-1}(u)\big). \tag{A2}$$

By assuming time-stationarity of the transformation $T$, it can be applied similarly in the projection period to link CDFs between the model and the reference:

$$T\big(F_{M_P}(x)\big) = F_{R_P}(x). \tag{A3}$$

By combining Eq. (A2) and Eq. (A3), we then can generate $F_{R_P}$, the estimated CDF of the climate variable in the reference during the projection period:

$$F_{R_P}(x) = F_{R_C}\left(F_{M_C}^{-1}(F_{M_P}(x))\right). \tag{A4}$$

Once $F_{R_P}$ has been estimated, a simple quantile-quantile method is performed between $F_{R_P}$ and $F_{M_P}$ to derive the bias corrected time series $\widehat{X}_{M_P}^d$ for the projection period as following:

$$\widehat{X}_{M_p}^d(t) = F_{R_P}^{-1}\left(F_{M_P}(X_{M_P}^d(t))\right). \tag{A5}$$

While a traditional quantile-mapping approach performed to correct a dataset $\mathbf{X}_{M_P}$ of simulations over the projection period will use the formulation $\widehat{X}_{M_p}^d(t) = F_{R_C}^{-1}\left(F_{M_C}(X_{M_P}^d(t))\right)$ (i.e., based on two distributions characterizing the calibration

period), the CDF-t method relies on Eq. (A5) where the two involved distributions characterize projected distributions. By proceeding this way, CDF-t takes into account the potential evolution of CDFs of the model between the calibration and projection periods to adjust the projection period. CDF-t is applied independently for each of the $D$ statistical dimensions and for both calibration and projection period to derive the final bias corrected outputs $\widehat{\mathbf{X}}_{M_C}$ and $\widehat{\mathbf{X}}_{M_P}$.

## Appendix B: Details on the R²D² method

The R²D² method, belonging to the "marginal/dependence" category, consists in several successive steps that are similar to adjust climate simulations for calibration and projection periods. Hence, to avoid redundancy, the correction procedure for the projection period will only be explained in this subsection. In this appendix, temporary corrected outputs for the projection period are denoted $\widetilde{\mathbf{X}}_{M_P}$.

– First, an univariate BC method is performed for the projection period to obtain the $N \times D$ matrix output $\widetilde{\mathbf{X}}_{M_P}$. As a

reminder, $\widetilde{\mathbf{X}}_{M_P} = \left[\left(\widetilde{X}_{M_P}^1(1),\ldots,\widetilde{X}_{M_P}^1(N)\right)',\ldots,\left(\widetilde{X}_{M_P}^D(1),\ldots,\widetilde{X}_{M_P}^D(N)\right)'\right]$;

– For each of the dimension $d$, R²D² computes the ranks of the time series within the univariate BC outputs $\widetilde{\mathbf{X}}_{M_P}$. For example, for the dimension $d$, the $N \times 1$ vector $\left(\text{rank}\left(\widetilde{X}_{M_P}^d(1)\right),\ldots,\text{rank}\left(\widetilde{X}_{M_P}^d(N)\right)\right)'$, denoted $\left(\widetilde{r}_{M_P}^d(1),\ldots,\widetilde{r}_{M_P}^d(N)\right)'$, is computed. It results in getting, for each time step $t$, a D-dimensional vector $\widetilde{\mathbf{R}}_{M_P}(t) = \left(\widetilde{r}_{M_P}^1(t),\ldots,\widetilde{r}_{M_P}^D(t)\right)$ which provides the multivariate rank structure of $\widetilde{\mathbf{X}}_{M_P}$ at $t$.

– For each of the dimension $d$, R²D² computes the ranks of the time series within the reference dataset during calibration $\mathbf{X}_{R_C}$. For example, for the dimension $d$, the $N \times 1$ vector $\left(\text{rank}\left(X_{R_C}^d(1)\right),\ldots,\text{rank}\left(X_{R_C}^d(N)\right)\right)'$, denoted $\left(r_{R_C}^d(1),\ldots,r_{R_C}^d(N)\right)'$, is computed. It results in getting, for each time step $t$, a D-dimensional vector $\mathbf{R}_{R_C}(t) = \left(r_{R_C}^1(t),\ldots,r_{R_C}^D(t)\right)$ which provides the multivariate rank structure of $\mathbf{X}_{R_C}$ at $t$.

– A "reference" dimension $d$ needs to be selected by the users in $\widetilde{\mathbf{X}}_{M_P}$. The corresponding univariate time series will be

kept untouched in the final R²D² outputs as the correction of the multivariate dependence structure is articulated on this dimension "pivot". For each time step $t$:

- the algorithm R$^2$D$^2$ finds $t^*$ such that $\widetilde{r}^d_{M_P}(t) = r^d_{R_C}(t^*)$. From $t^*$, R$^2$D$^2$ deduces the multivariate rank structure of the reference during the calibration period at this specific time step: $\mathbf{R}_{R_C}(t^*) = \left(r^1_{R_C}(t^*),\ldots,r^D_{R_C}(t^*)\right)$;

- R$^2$D$^2$ forces the $D$-dimensional vector of ranks of its final outputs $\widehat{\mathbf{X}}_{M_P}$ to be equal to:

$$\widehat{\mathbf{R}}_{M_P}(t) = \left(r^1_{R_C}(t^*),\ldots,\widetilde{r}^d_{M_P}(t),\ldots,r^D_{R_C}(t^*)\right).$$

To do so, the algorithm looks for shuffling the values in each of the dimensions $k \neq d$ of $\widetilde{\mathbf{X}}_{M_P}$, such that its rank structure at time $t$ matches $\widehat{\mathbf{R}}_{M_P}(t)$. In a more explicit way, for all $k \neq d$, R$^2$D$^2$ finds the time steps $t_k$ such that $r^k_{R_C}(t^*) = \widetilde{r}^k_{M_P}(t_k)$. The value in $\widetilde{X}^k_{M_P}$ to shuffle associated with the rank $\widetilde{r}^k_{M_P}(t_k)$ is then derived, and copied in the final outputs $\widehat{X}^k_{M_P}(t)$.

- By repeating the step 4 until each dimension has been used one time as a "reference" for the shuffling, R$^2$D$^2$ is able to derive a collection of $D$ MBC outputs, with exactly the same multivariate dependence structure but differing on temporal properties, describing the possible variability in the different rank structures.

## Appendix C: Details on the dOTC method

dOTC method, belonging to the "all-in-one" category, relies on optimal transport theory to adjust climate simulations. A slight different mathematical notation needs to be used here to explain dOTC. Let define $\mathbf{X}_{R_C}(t)$ the realizations of $\mathbf{X}_{R_C}$ at each time step $t$ across each of the $D$ dimensions. The collection of the variables $\left(\mathbf{X}_{R_C}(1),\ldots,\mathbf{X}_{R_C}(N)\right)$ forms a $D \times N$ matrix, and describes $\mathbf{X}_{R_C}$ in a different way. Similarly, $\left(\mathbf{X}_{M_C}(1),\ldots,\mathbf{X}_{M_C}(N)\right)$ and $\left(\mathbf{X}_{M_P}(1),\ldots,\mathbf{X}_{M_P}(N)\right)$ are considered for, respectively, $\mathbf{X}_{M_C}$ and $\mathbf{X}_{M_P}$. In the following, $\mathbf{c}_i$ denotes a collection of multivariate cells that partition regularly $\mathbb{R}^D$ and fully cover $\left(\mathbf{X}_{M_C}(1),\ldots,\mathbf{X}_{M_C}(N)\right)$ and $\left(\mathbf{X}_{M_P}(1),\ldots,\mathbf{X}_{M_P}(N)\right)$. To simplify notations, the center of a grid cell $\mathbf{c}_i$ is also denoted $\mathbf{c}_i$. Hereinafter is presented first how dOTC adjusts the calibration period of climate simulations to derive $\widehat{\mathbf{X}}_{M_C}$. Then, the algorithm procedure will be detailed for the adjustment of the projection period $\widehat{\mathbf{X}}_{M_P}$.

**The "OTC" procedure for the calibration period:**

- First, the algorithm estimates $\widetilde{\mathbb{P}}_{\mathbf{X}_{R_C}}$ and $\widetilde{\mathbb{P}}_{\mathbf{X}_{M_C}}$ the empirical multivariate distributions of $\mathbf{X}_{R_C}$ and $\mathbf{X}_{M_C}$. To do so, dOTC computes a sum of Dirac masses. For example, for $\mathbf{X}_{M_C}$, we have:

$$\widetilde{\mathbb{P}}_{\mathbf{X}_{M_C}}(A) = \sum_{i=1}^{I} p_{\mathbf{X}_{M_C},i}\,\delta_{\mathbf{c}_i}(A),$$

where $p_{\mathbf{X}_{M_C},i} = \frac{1}{N}\sum_{t=1}^{N} \mathbf{1}(\mathbf{X}_{M_C}(t) \in \mathbf{c}_i)$, and $A \subset \mathbb{R}^D$.

- Then, the coefficients $\gamma_{ij}$ defining the estimator $\widetilde{\gamma}$ of the optimal plan that moves the bin $\mathbf{c}_i$ of $\widetilde{\mathbb{P}}_{\mathbf{X}_{M_C}}$ to the bin $\mathbf{c}_j$ of $\widetilde{\mathbb{P}}_{\mathbf{X}_{R_C}}$ are computed. For $A, B \subset \mathbb{R}^D$, $\widetilde{\gamma}$ is defined as follows:

$$\widetilde{\gamma}(A \times B) = \sum_{i,j=1}^{I,J} \gamma_{ij} \delta_{(\mathbf{c}_i, \mathbf{c}_j)}(A \times B),$$

The coefficient $\gamma_{ij}$ corresponds to the joint probability of $\mathbf{X}_{M_C}$ being in $\mathbf{c}_i$ and $\mathbf{X}_{R_C}$ being in $\mathbf{c}_j$, which is part of the MBC process. They have to respect the following constraints:

$$\sum_{j=1}^{J} \gamma_{ij} = p_{\mathbf{X}_{M_C},i},$$

$$\sum_{i=1}^{I} \gamma_{ij} = p_{\mathbf{X}_{R_C},j},$$

and have to minimize the following cost function $\widetilde{C}$:

$$\widetilde{C}(\widetilde{\gamma}) = \sum_{i,j=1}^{I,J} \|\mathbf{c}_i - \mathbf{c}_j\|^2 \gamma_{ij}.$$

To find these coefficients that form the so-called optimal transport plan, the algorithm resolves the linear programming problem by using the procedure developed by Flamary and Courty (2017).

– Then, for each time step $t$:

    – the algorithm finds the cell $\mathbf{c}_i$ containing $\mathbf{X}_{M_C}(t)$

    – Using the plan $\gamma_{ij}$, it constructs the conditional probability vector $\widetilde{\gamma}_{\mathbf{X}_{M_C}(t)} = (\gamma_{i,1}, \ldots, \gamma_{i,J})/p_{\mathbf{X}_{M_C},i}$

    – According to the probability vector $\widetilde{\gamma}$, the algorithm draws a $j^* \in 1, \ldots, J$

    – The correction $\widehat{\mathbf{X}}_{M_C}(t)$ is then derived with an uniform draw in $\mathbf{c}_{j^*}$

– After iterating for each $t$, the final outputs for the calibration period $\widehat{\mathbf{X}}_{M_C}$ is obtained.

**The "dOTC" procedure for the projection period**

– As explained before, dOTC estimates $\widetilde{\mathbb{P}}_{\mathbf{X}_{R_C}}$, $\widetilde{\mathbb{P}}_{\mathbf{X}_{M_C}}$ and $\widetilde{\mathbb{P}}_{\mathbf{X}_{M_P}}$ the empirical multivariate distributions of $\mathbf{X}_{R_C}$, $\mathbf{X}_{M_C}$ and $\mathbf{X}_{M_P}$.

– Then, the coefficients $\gamma_{ij}$ defining the estimator $\widetilde{\gamma}$ of the optimal plan that moves the bin $\mathbf{c}_i$ of $\widetilde{\mathbb{P}}_{\mathbf{X}_{M_C}}$ to the bin $\mathbf{c}_j$ of $\widetilde{\mathbb{P}}_{\mathbf{X}_{R_C}}$ are computed.

- Similarly, the coefficients $\varphi_{ik}$ defining the estimator $\widetilde{\varphi}$ of the optimal plan that moves the bin $\mathbf{c}_i$ of $\widetilde{\mathbb{P}}_{\mathbf{X}_{M_C}}$ to the bin $\mathbf{c}_k$ of $\widetilde{\mathbb{P}}_{\mathbf{X}_{M_P}}$ are computed.

- By default, the diagonal matrix of the standard deviations $\mathbf{D}$ is computed: $\mathbf{D} = \mathrm{diag}(\sigma_{X_{M_C}} \sigma_{X_{R_C}}^{-1})$. Others alternatives for the computation of $\mathbf{D}$ are possible and detailed in Robin et al. (2019).

- Then, for each time step $t$:

  - the algorithm finds the cell $\mathbf{c}_j$ containing $\mathbf{X}_{R_C}(t)$,

  - Using the plan $\gamma_{ij}$, it finds the cell $\mathbf{c}_i$ of $\widetilde{\mathbb{P}}_{\mathbf{X}_{M_C}}$ associated with $\mathbf{c}_j$

  - Using the plan $\varphi_{ik}$, it finds the cell $\mathbf{c}_k$ of $\widetilde{\mathbb{P}}_{\mathbf{X}_{M_P}}$ associated with $\mathbf{c}_i$

  - Using $\mathbf{D}$, it computes the vector $\mathbf{v}_{ik} := \mathbf{c}_k - \mathbf{c}_i$ for scaling adjustment of the correction

  - A preliminary (and temporary) correction of the model during the projection $\check{\mathbf{X}}_{M_P}(t)$ is then obtained: $\check{\mathbf{X}}_{M_P}(t) = \mathbf{X}_{M_C}(t) + \mathbf{D}.\mathbf{v}_{ik}$

- Then, it estimates $\check{\mathbb{P}}_{\check{\mathbf{X}}_{M_P}}$ the empirical multivariate distribution of $\check{\mathbf{X}}_{M_P}$.

- Finally, the OTC procedure (see above for calibration period) is applied between $\left(\mathbf{X}_{M_P}(1),\ldots,\mathbf{X}_{M_P}(N)\right)$ and $\left(\check{\mathbf{X}}_{M_P}(1),\ldots,\check{\mathbf{X}}_{M_P}(N)\right)$ to produce the final outputs $\left(\widehat{\mathbf{X}}_{M_P}(1),\ldots,\widehat{\mathbf{X}}_{M_P}(N)\right)$.

## Appendix D: Details on the MBCn method

The MBCn method can be summarized in 3 steps in the way it corrects climate simulations. As a reminder, MBCn belongs to the "marginal/dependence" category, i.e. correcting separately marginal distributions and full dependence structure of climate simulations. In this appendix, temporary corrected outputs of a matrix $\mathbf{X}_A$ are denoted with tilde accents ($\widetilde{\mathbf{X}}_A$) or inverted hats ($\check{\mathbf{X}}_A$).

- First, marginal distributions are corrected with an univariate BC method. To do so, MBCn uses the Quantile Delta Mapping (QDM from Cannon et al., 2015) algorithm defined as follows:

$$\begin{cases} \widetilde{X}^d_{M_C}(t) & = F_{R_C}^{-1}\left( F_{M_C}\left( X^d_{M_C}(t) \right) \right) \\ \Delta_t & = X^d_{M_P}(t) - F_{M_C}^{-1}\left( F_{M_P}\left( X^d_{M_P}(t) \right) \right) \\ \widetilde{X}^d_{M_P}(t) & = F_{R_C}^{-1}\left( F_{M_P}\left( X^d_{M_P}(t) \right) \right) + \Delta_t \end{cases} \tag{D1}$$

This transfer function preserves absolute changes in quantiles and has to be applied for interval variables such as temperature. For ratio variables like precipitation, the addition/substraction operators in the transfer function has to be replaced

by multiplication/division operators to define a function that preserves relative changes in quantiles. For both calibration and projection period, the $D$ physical variables are independently adjusted by applying the corresponding transfer function. The resulting matrices $\widetilde{\mathbf{X}}_{M_C}$ and $\widetilde{\mathbf{X}}_{M_P}$ with adjusted marginal distributions are stored by the algorithm in respectively $\widetilde{\mathbf{X}}_{M_C}^{init}$ and $\widetilde{\mathbf{X}}_{M_P}^{init}$ before the second step, as it re-uses them in the third one.

– Within the MBCn algorithm, the multivariate dependence structure of the simulations is adjusted through an iterative procedure. At each iteration $j$, an application of a $D \times D$ random orthogonal rotation matrix $\mathbf{R}^{[j]}$ (Mezzadri, 2007) is performed on the datasets $\mathbf{X}_{R_C}$, $\widetilde{\mathbf{X}}_{M_C}$ and $\widetilde{\mathbf{X}}_{M_P}$ obtained from Step 1:

$$\begin{cases} \check{\mathbf{X}}_{R_C}^{[j]} & = \mathbf{X}_{R_C}^{[j]} \mathbf{R}^{[j]} \\ \check{\mathbf{X}}_{M_C}^{[j]} & = \widetilde{\mathbf{X}}_{M_C}^{[j]} \mathbf{R}^{[j]} \\ \check{\mathbf{X}}_{M_P}^{[j]} & = \widetilde{\mathbf{X}}_{M_P}^{[j]} \mathbf{R}^{[j]} \end{cases} \tag{D2}$$

It permits to provide linear combinations of the original variables. The QDM transfer function defined in Eq. (D1) for interval variables, i.e. with addition/substraction operators, is then applied on each of the rotated marginal distributions of $\check{\mathbf{X}}_{M_C}^{[j]}$ and $\check{\mathbf{X}}_{M_P}^{[j]}$, considering the corresponding rotated marginal distributions in $\check{\mathbf{X}}_{R_C}^{[j]}$ as the reference. Once marginal distributions have been adjusted in $\check{\mathbf{X}}_{M_C}^{[j]}$ and $\check{\mathbf{X}}_{M_P}^{[j]}$, matrices are rotated back to the physical variables ranges:

$$\begin{cases} \mathbf{X}_{R_C}^{[j+1]} & = \mathbf{X}_{R_C}^{[j]} \\ \widetilde{\mathbf{X}}_{M_C}^{[j+1]} & = \check{\mathbf{X}}_{M_C}^{[j]} R^{[j]^{-1}} \\ \widetilde{\mathbf{X}}_{M_P}^{[j+1]} & = \check{\mathbf{X}}_{M_P}^{[j]} R^{[j]^{-1}} \end{cases} \tag{D3}$$

These successive steps are applied iteratively until the multivariate distribution of the corrected simulations $\widetilde{\mathbf{X}}_{M_C}^{[j+1]}$ matches the one of the reference $\mathbf{X}_{R_C}$.

– Once the full dependence structure of simulated variables converged to the one of the reference after, let say, the $j^*$-th iteration, MBCn replaces quantiles of each of the variables in $\widetilde{\mathbf{X}}_{M_C}^{[j^*+1]}$ and $\widetilde{\mathbf{X}}_{M_P}^{[j^*+1]}$ obtained at the end of Step 2) with those from $\widetilde{\mathbf{X}}_{M_C}^{init}$ and $\widetilde{\mathbf{X}}_{M_P}^{init}$ obtained during Step 1). This additional step preserves from the possible deterioration of the model trend during the correction of the multivariate dependence structure in Step 2). Simulations with corrected marginal distributions features and full dependence structure $\widehat{\mathbf{X}}_{M_C}$ and $\widehat{\mathbf{X}}_{M_P}$ are then obtained.

## Appendix E: Details on the MRec method

The MRec method, belonging to the "all-in-one" category, consists in the following steps.

– First, each of the D dimensions in $\mathbf{X}_{R_C}$ is transformed independently in the Gauss domain. However, the transformation differs between interval variables, i.e. temperature, and ratio variables, i.e. precipitation, and is performed as follows:

      – For a dimension $d$ being an interval variable, a distribution $F_{R_C}^d$ is fitted:

$$F_{R_C}^d(x) = \mathbb{P}\big(X_{R_C}^d(t) < x\big).$$

Then, the corresponding vector $W^d$ is computed as following:

$$W^d(t) = \Phi^{-1}\Big(F_{R_C}^d\big(X_{R_C}^d(t)\big)\Big),$$

with $\Phi$ the distribution function of the standard normal distribution $\mathcal{N}(0,1)$.

      – For a dimension $k$ being a ratio variable, a distribution $F_{R_C}^k$ is fitted:

$$F_{R_C}^k(x) = \mathbb{P}\big(X_{R_C}^k(t) < x | X_{R_C}^k(t) > 0\big).$$

Additionally, the frequency $P_{k0}$ of null events in $X_{R_C}^k$ is computed:

$$P_{k0} = \mathbb{P}\big(X_{R_C}^k(t) = 0\big).$$

Then, the corresponding vector $W^k$ is computed as following:

$$W^k(t) = \begin{cases} \Phi^{-1}\Big(F_{R_C}^k\big(X_{R_C}^k(t)\big)(1 - P_{k0}) + P_{k0}\Big) \\ \Phi^{-1}\Big(\frac{P_{k0}}{2}\Big). \end{cases}$$

Doing this step for each dimension permits to derive the matrix $\mathbf{W}$ of dimension $N \times D$, composed of the Gaussian transformed vectors $W^1, \ldots, W^D$.

Following the notation in Bárdossy and Pegram (2012), the same procedure is repeated for $\mathbf{X}_{M_C}$ and $\mathbf{X}_{M_P}$ to derive respectively the Gaussian transformed data $\mathbf{Y}$ and $\mathbf{Y}'$.

   – For both Gaussian transformed data $\mathbf{W}$ and $\mathbf{Y}$, the $N \times N$ Pearson cross correlation matrices $\mathbf{C}_W$ and $\mathbf{C}_Y$ are computed.

   – A Singular Value Decomposition (SVD) is applied on $\mathbf{C}_W$ such that:

$$\mathbf{C}_W = \mathbf{A}_W \mathbf{D}_W \mathbf{B}_W^T,$$

with $\mathbf{A}_W$ and $\mathbf{B}_W$ having same dimensions as $\mathbf{C}_W$, and $\mathbf{D}_W$ a diagonal matrix of singular values. From this decomposition, the square root matrix of $\mathbf{C}_W$, denoted $\mathbf{S}_W$, can be obtained as followed:

$$\mathbf{S}_W = \mathbf{A}_W \mathbf{D}_W^{1/2} \mathbf{A}_W^T.$$

- Similarly, a Singular Value Decomposition (SVD) is applied on $\mathbf{C}_Y$ such that:

$$\mathbf{C}_Y = \mathbf{A}_Y \mathbf{D}_Y \mathbf{B}_Y^T.$$

From this decomposition, its inverse square root matrix $\mathbf{T}_Y$ can be obtained as following:

$$\mathbf{T}_Y = \mathbf{A}_Y \mathbf{D}_Y^{-1/2} \mathbf{A}_Y^T.$$

- $\mathbf{Y}$ is decorrelated to $\mathbf{Q}$: $\mathbf{Q} = \mathbf{Y}\mathbf{T}_Y$.

- $\mathbf{Q}$ is then recorrelated to $\mathbf{V}$: $\mathbf{V} = \mathbf{Q}\mathbf{S}_W$. $\mathbf{V}$ is hence the recorrelated transformed model data for the calibration period presenting the same correlation structure as $\mathbf{W}$.

- For the projection period, $\mathbf{V}'$ is computed directly without decorrelation step: $\mathbf{V}' = \mathbf{Y}'\mathbf{T}_Y\mathbf{S}_W$.

- $\mathbf{V}$ and $\mathbf{V}'$ are then transformed back to physical variables using univariate quantile-quantile method for each dimension $d$, with $\mathbf{X}_{R_C}^d$ being the target for the correction. The desired adjusted matrices $\widehat{\mathbf{X}}_{M_C}$ and $\widehat{\mathbf{X}}_{M_P}$ are then finally obtained.

*Code and data availability.* The R package for MBCn is available on https://CRAN.R-project.org/package=MBC. $R^2D^2$ is not publicly available yet but only on demand to M. Vrac. dOTC and MRec are publicly available on https://github.com/yrobink/SBCK.

*Author contributions.* MV had the initial idea of the study. MV and BF designed the experiments and protocols. BF made all computations and figures. BF and MV made the analyses and interpretations. They also wrote the first draft of the article and iterations on the writting were made with all co-autors.

*Competing interests.* The authors declare that they have no competing interests.

*Acknowledgements.* BF and MV acknowledge financial support from the EUPHEME project. MV also acknowledges support from the
CoCliServ project and French "Convention de Service Climatique". Both EUPHEME and CoCliServ are part of ERA4CS, an ERA-NET initiated by JPI Climate and co-funded by the European Union (grant no. 690462). This work was supported by the metaprogramme Adaptation of Agriculture and Forest to Climate Change (AAFCC) of the French National Research Institute for Agriculture, Food Environment (INRAE).

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

**Table 1.** Summary of attributes of the different bias-correction methods.

| Characteristics | CDF-t | $R^2D^2$ | dOTC | MBCn | MRec |
|---|---|---|---|---|---|
| Type of BC | 1d-BC | MBC | MBC | MBC | MBC |
| Category of MBC | n.a. | Marginal/dependence | All-in-one | Marginal/dependence | All-in-one |
| Statistical technique | Non-stationary quantile mapping | Conditionnal resampling | Optimal transport | Iterative partial matrix recorrelation | Matrix recorrelation |
| Dependence structure | ∼ same as the model | ∼ same as the reference | Allows changes in the dep. struct. | Allows changes in the dep. struct. | Allows changes in the *Gaussian* dep. struct. |
| Conceptual feature | Deterministic | Deterministic and stochastic | Stochastic | Deterministic and stochastic | Deterministic |

**Table 2.** Summary of recommendations for the multivariate BC methods to use with respect to the different assumptions made by practitioners or end-users. Green checks and red crosses indicate whether BC methods are recommended for use or not, depending on the statement in rows. Orange tildes indicate when particular caution has to be taken.

| Characteristics | CDF-t | $R^2D^2$ | dOTC | MBCn | MRec |
|---|---|---|---|---|---|
| Correction of univariate distrib. prop. | ✓ | ✓ | ✓ | ✓ | ✓ |
| Modification of the correlations of the model | ✗ | ✓ | ✓ | ✓ | ✓ |
| Capacity to correct inter-var. prop. | ✗ | ✓ | ✓ | ✓ | ✓ |
| Capacity to correct spatial prop. | ✗ | ✓ | ✓ | ∼ | ∼ |
| Capacity to correct temporal prop. | ✗ | ✗ | ✗ | ✗ | ✗ |
| Preserve the rank structure of the model | ✓ | ∼ | ∼ | ∼ | ∼ |
| Capacity to correct small geographical area | n.a. | ✓ | ✓ | ✓ | ✓ |
| Capacity to correct large geographical area | n.a. | ∼ | ∼ | ∼ | ✗ |
| Allow for evolution of the rank dependence | ✓ | ✗ | ✓ | ∼ | ✓ |

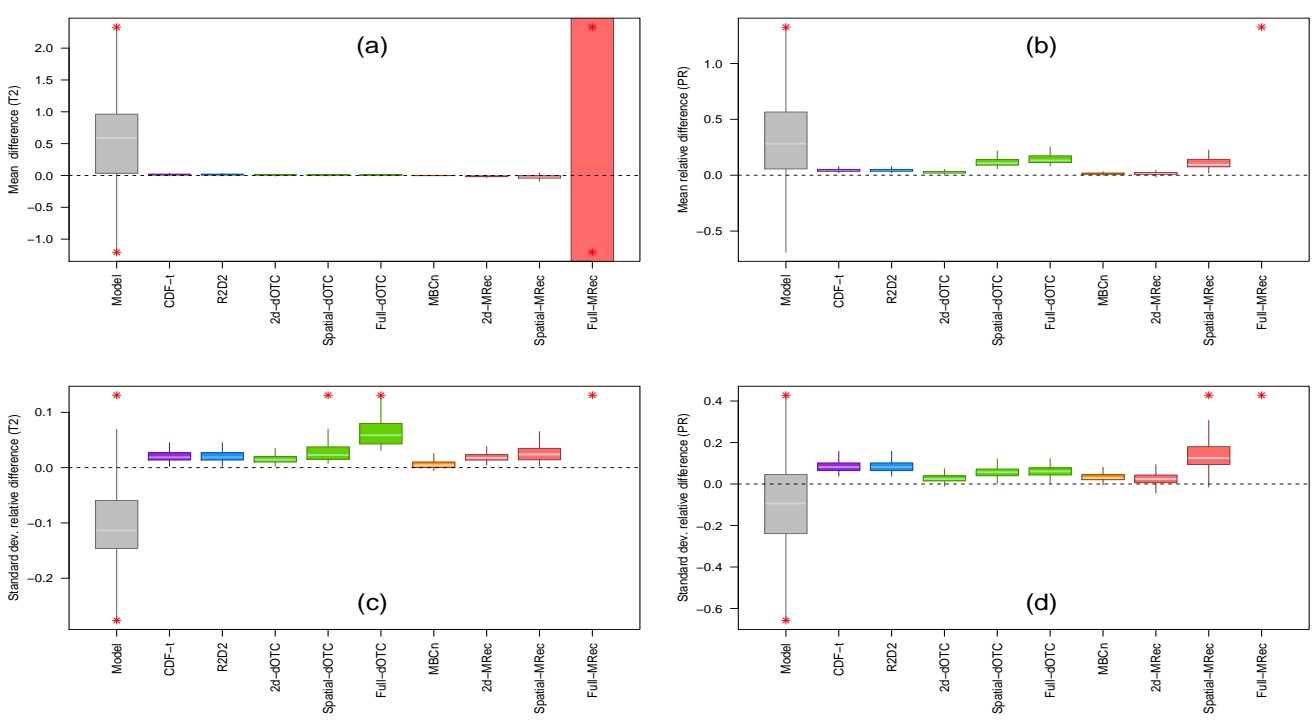

**Figure 1.** Boxplots of mean **(a and b)** and standard deviation **(c and d)** differences for Temperature (T2, a and c) and Precipitation (PR, b and d) during winter over the 1979-2016 period for France (WFDEI reference). Results are shown for: plain IPSL; CDF-t; $R^2D^2$; dOTC (2d-, Spatial- and Full-versions); MBC-n and MRec (2d-, Spatial- and Full-versions) outputs. Red asterisks indicate values lying outside the plotted range.

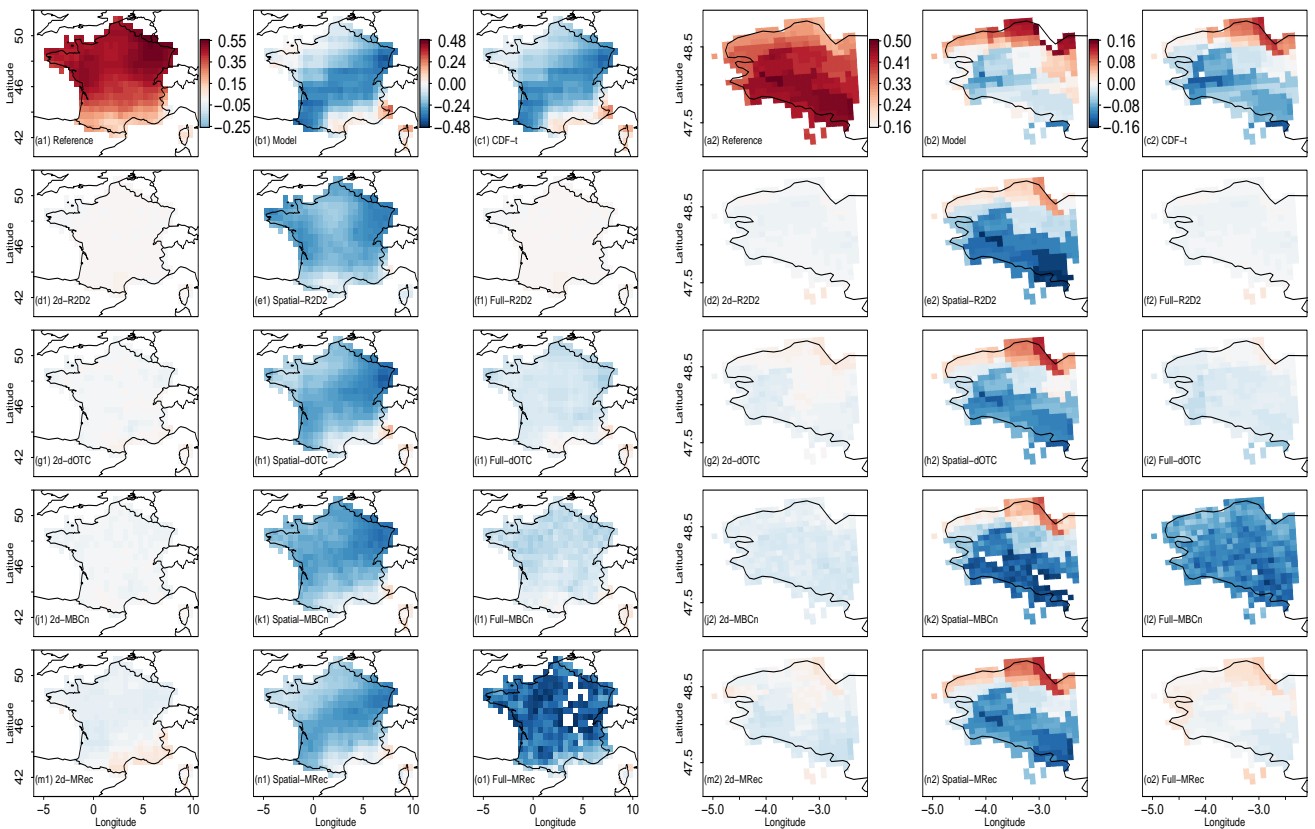

**Figure 2.** Differences of temperature vs. precipitation Spearman correlation computed at each grid cell for BC methods using WFDEI reference **(a1-o1)** and SAFRAN reference **(a2-o2)** during winter over the 1979-2016 period. Results are shown for: Reference; plain IPSL; CDF-t; $R^2D^2$; dOTC; MBC-n and MRec outputs for respectively 2d-, Spatial- and Full- versions. Note that the color scales between **(a1-o1)** and **(a2-o2)** are not the same to better emphasize intensities of values in the two regions.

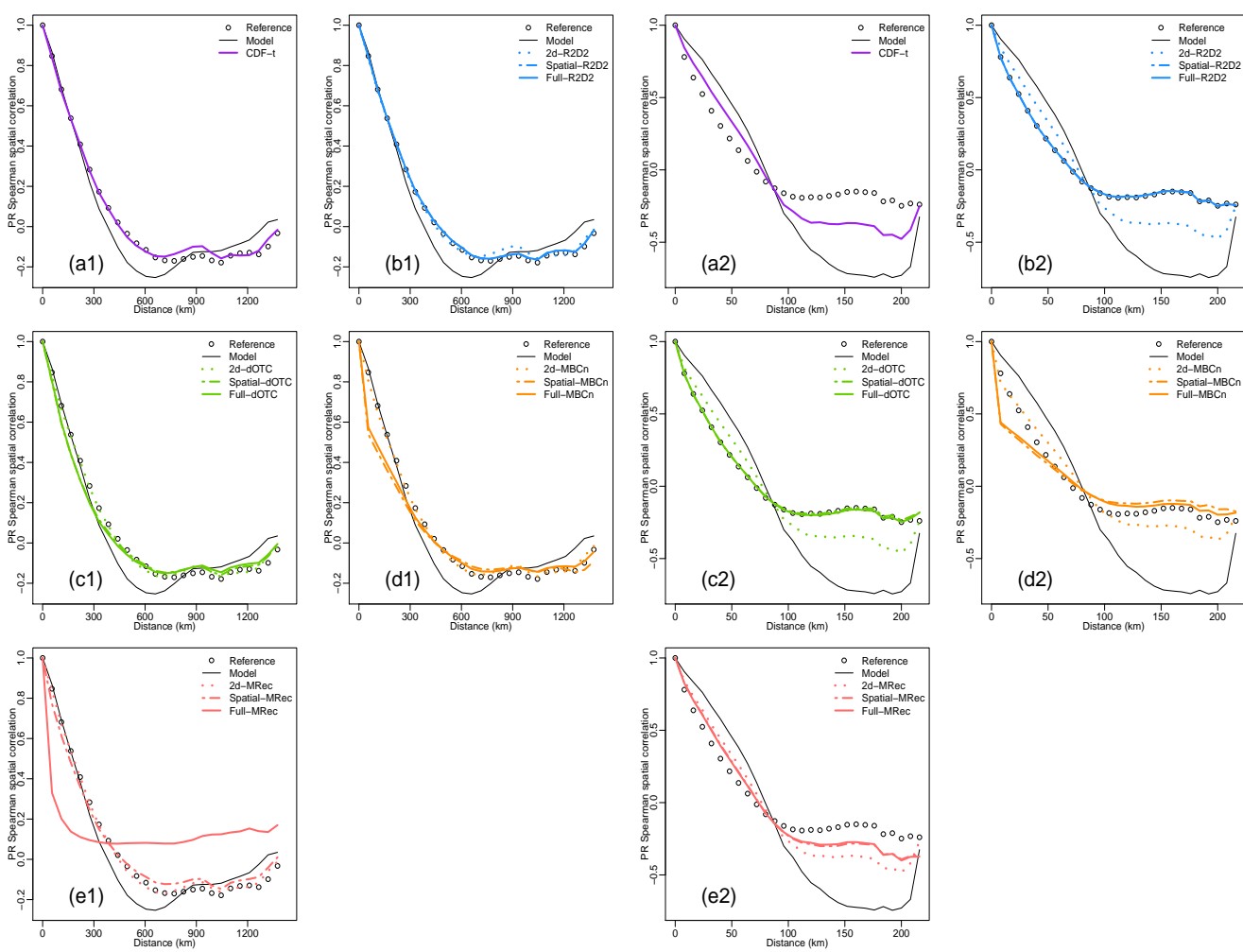

**Figure 3.** Correlograms for precipitation using WFDEI reference for France **(a1-e1)** and SAFRAN reference for Brittany **(a2-e2)** during winter over the 1979-2016 period. Results are shown for Reference (circles) and plain IPSL (black line). Results are displayed for: CDF-t; $R^2D^2$; dOTC; MBC-n and MRec outputs for respectively 2d- (dotted), Spatial- (dashed) and Full-versions (solid lines).

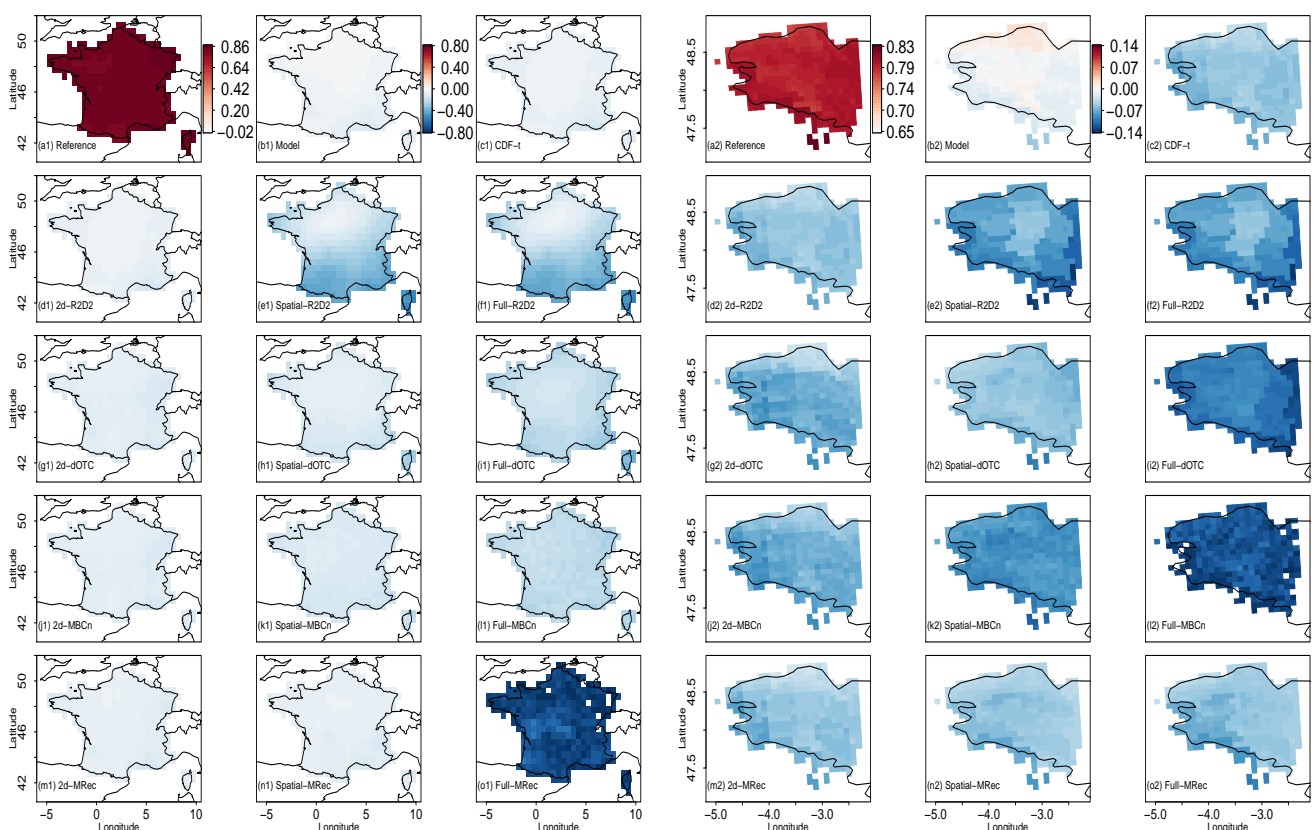

**Figure 4.** Differences of order 1 Pearson autocorrelation for temperature using WFDEI reference **(a1-o1)** and SAFRAN reference **(a2-o2)** during winter over the 1979-2016 period. Results are shown for: Reference, plain IPSL, CDF-t, $R^2D^2$, dOTC, MBC-n and MRec outputs for 2d-, Spatial- and Full-versions. Note that the color scales between **(a1-o1)** and **(a2-o2)** are not the same to better emphasize intensities of values of the two regions.

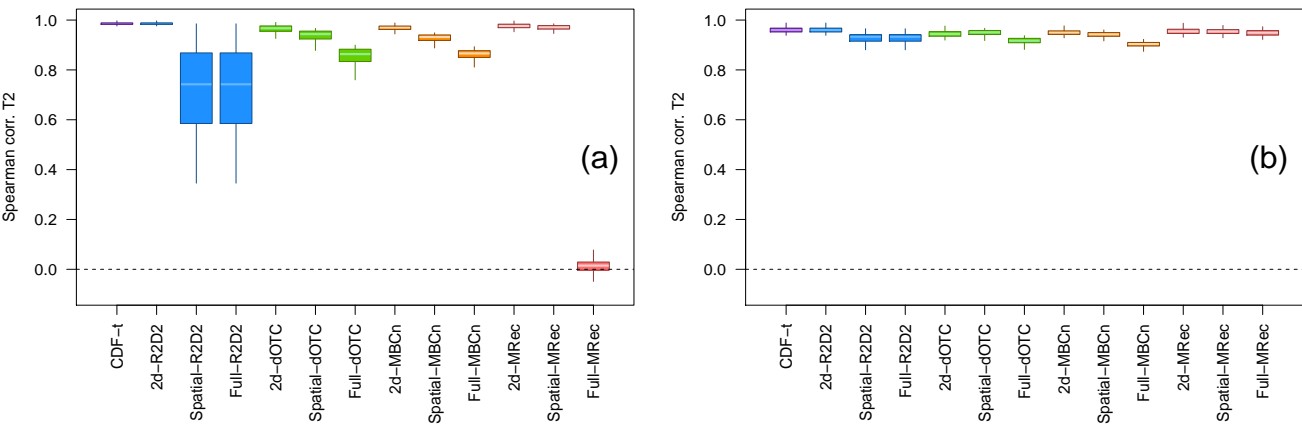

**Figure 5.** Boxplots of rank correlations computed at each grid cell between the bias corrected and the raw climate model time series, for temperature, using WFDEI for France **(a)** and SAFRAN for Brittany **(b)** region during winter over the 1979-2016 period. Results are shown for: CDF-t, $R^2D^2$, dOTC, MBC-n and MRec outputs for 2d-, Spatial- and Full-versions.

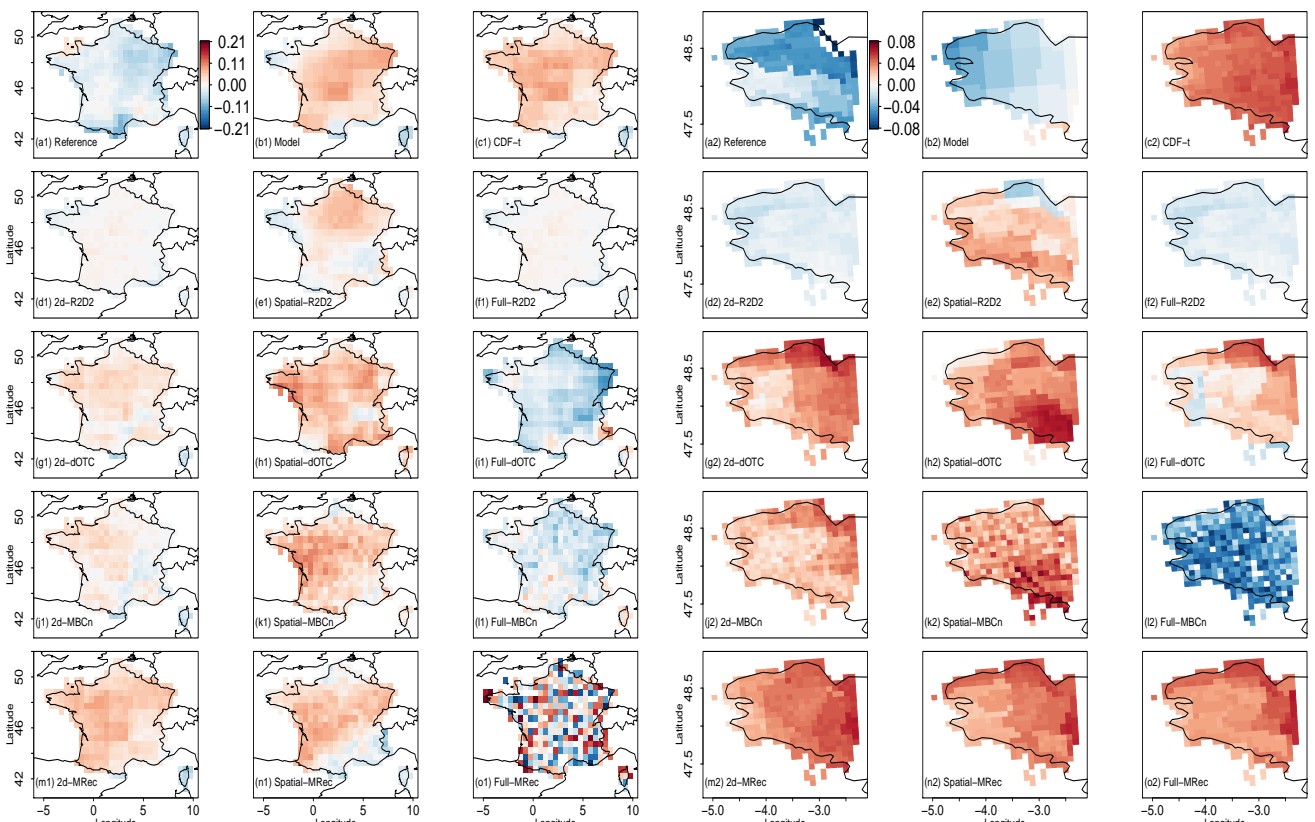

**Figure 6.** Differences of temperature vs. precipitation Spearman correlations computed at each grid cell between the 1979-1997 and 1998-2016 periods during winter. WFDEI **(a1-o1)** and SAFRAN **(a2-o2)** data are used as references for the bias correction. Note that the color scales between **(a1-o1)** and **(a2-o2)** are not the same to better emphasize intensities of values of the two regions.

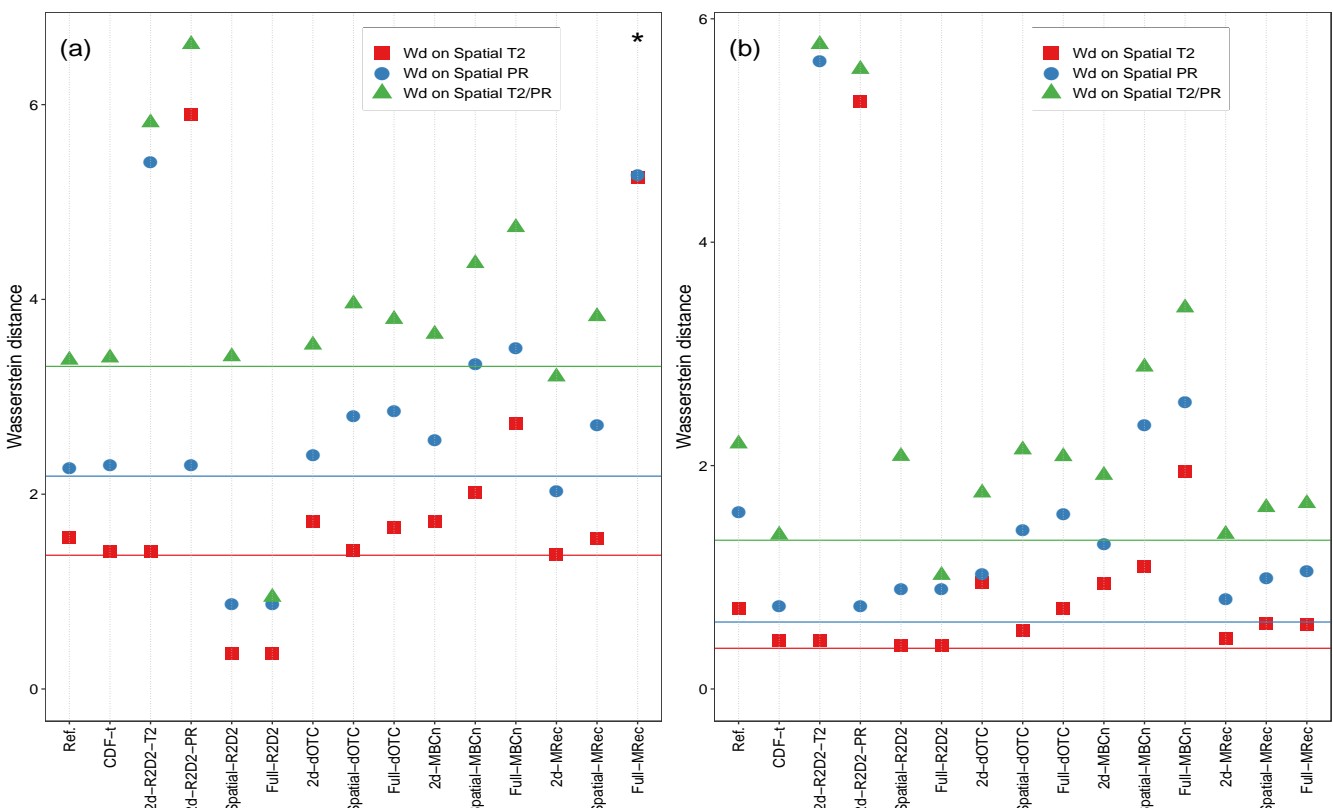

**Figure 7.** Values of the three Wasserstein distances on ranks between 1979-1997 and 1998-2016 periods during winter for temperature (square), precipitation (circle) and both temperature and precipitation (triangle) for the region of France **(a)** and Brittany **(b)**. Results are presented for the reference, plain IPSL (lines); CDF-t and the different MBCs. 2d-$R^2D^2$-T2 (resp. 2d-$R^2D^2$-PR) indicates results for 2d-$R^2D^2$ with temperature (resp. precipitation) used as reference dimension. Black asterisks indicate values lying outside the plotted range.