# Peer review of "Multivariate bias corrections of climate simulations: Which benefits for which losses?"

_Earth System Dynamics, 2020_

## Referee Comment (RC1) · Jakob Zscheischler (Referee) · 26 Mar 2020

This is a timely paper providing an overview about the plethora of newly emerging multivariate bias correction approaches that have been developed over the recent years. The authors provide recommendation about which approach should be used under which conditions. The paper has the potential to become a key reference for multivariate bias correction approaches. It has an easy to follow clear structure, is well written and falls into the scope of ESD. I have a few minor recommendations which should help to improve its accessibility and impact.

Introduction: I miss some strong arguments why and in which situations we need MBC. For many impacts, univariate BC is (probably) enough and MBC does not provide

are large boost in performance. Indeed a number of studies have argued over the last years that for their application domain MBC does not outperform univariate BC (Yang et al., 2015; Casanueva et al., 2018; Räty et al., 2018). However, I would argue that these results cannot be generalized. One particularly relevant field of application where MBC should be highly beneficial is the area of compound events, where multiple climate drivers result in a large impact (Zscheischler et al., 2018). Arguably, a bias in the dependence structure of the drivers can result in unknown biases of the modelled impacts, which may even be aggravated by univariate BC (Zscheischler et al., 2019).

I'm not sure I entirely agree with the interpretation of Section 5.5.2 and figure 7. As I understand it, Wd only measures a distance. Hence if one obtains a similar value >0 it is unclear whether the change goes into the same direction. One might obtain a similar value for Wd but very different changes in the underlying distributions (though I admit that this would be coincidence and might not be very likely). I think this caveat should be mentioned.

L 604: Other examples for changes in dependence that might be highly relevant for impacts are: - increases in the dependence between storm surge and heavy precipitation in US coasts in the historical period (Wahl et al., 2015): affects the risk of compound floods; - increase in the strength of dependence between seasonal summer temperature and precipitation of most land regions with increasing warming (Zscheischler & Seneviratne, 2017): affects the likelihood of compound hot and dry events with a large array of impacts

L 642: This is easier said than done. The largest challenge in evaluating impact modelling output is the availability of impact data. It will therefore be difficult to decide which BC approach is more appropriate. That said, I agree that creating an ensemble of different approaches might help to cover uncertainties that are not only related to the choice of the GCM and forcing scenario but also the choice of BC method.

Figure 2 and 4: The correlations could be plotted as difference to the reference to

highlight the differences.

References:

Casanueva, A., Bedia, J., Herrera, S., Fernández, J., and Gutiérrez, J. M.: Direct and component-wise bias correction of multi-variate climate indices: the percentile adjustment function diagnostic tool, Climatic Change, 147, 411–425, https://doi.org/10.1007/s10584-018-2167-5, 2018.

Räty, O., Räisänen, J., Bosshard, T., and Donnelly, C.: Intercompar- ison of Univariate and Joint Bias Correction Methods in Changing Climate From a Hydrological Perspective, Climate, 6, 33, https://doi.org/10.3390/cli6020033, 2018.

Yang, W., Gardelin, M., Olsson, J., and Bosshard, T.: Multi-variable bias correction: application of forest fire risk in present and future climate in Sweden, Nat. Hazards Earth Syst. Sci., 15, 2037–2057, https://doi.org/10.5194/nhess-15-2037-2015, 2015.

Wahl, T., S. Jain, J. Bender, S. D. Meyers, and M. E. Luther: Increasing risk of compound flooding from storm surge and rainfall for major US cities. Nature Climate Change, 5 , doi: 10.1038/nclimate2736, 2015.

Zscheischler, J., Westra, S., van den Hurk, B. J. J., Pitman, A., Ward, P., Bresch, D. N., Leonard, M., Zhang, X., AghaKouchak, A., Wahl, T., and Seneviratne, S. I.: Future climate risk from compound events, Nat. Clim. Change, 8, 469–477, https://doi.org/10.1038/s41558-018-0156-3, 2018.

Zscheischler, J., Fischer, E. M., and Lange, S.: The effect of univariate bias adjustment on multivariate hazard estimates, Earth Syst. Dynam., 10, 31–43, https://doi.org/10.5194/esd-10-31-2019, 2019.

Zscheischler, J. and Seneviratne, S.
.: Dependence of drivers affects risks associated with compound events, Science Advances, 3, e1700263, https://doi.org/10.1126/sciadv.1700263, 2017.

---

## Referee Comment (RC2) · Anonymous Referee #2 · 11 Apr 2020

General comments:

1) A comparison of methods is especially helpful in an emerging area of research such as multivariate bias correction (MBC), where few guidelines are available. Overall, I like the article but have some suggestions for improvement.

2) I was surprised that the authors re-gridded the 0.5 degree precipitation product to the coarser climate model grid using nearest neighbor interpolation. The authors don't say what method is used for the 8km precipitation product, but presumably that also is nearest neighbor? Overall, this approach seems to ignore a lot of spatial information in the "observed" data and effectively makes this a quasi-regular resampling of the observed data and not an interpolation. Is the goal to get area-averaged precipitation or a gridded product of point precipitation? Some discussion of this choice and the

tradeoffs is warranted, unless the authors decide to use a different approach.

3) A lagged version of a variable, whether spatial or temporal, can be thought of as just another variable. Adding a lagged variable to MBC, therefore, is just MBC. As a result, I think the results that show the impacts of poorly conditioned matrices is the key takeaway here. One should parsimoniously add variables that are important to preserving the kind of variability that one is most interested in. For instance, if heatwaves are of interest, then one should emphasize temporal correlation. I'd like to see a bit more on the tradeoffs between emphasizing temporal vs spatial correlation, in terms of choosing the dimensionality of the bias correction. The authors touch on this in the discussion, but some more specific guidance for making these choices would be helpful.

4) Given that the methods use covariance (which is the basis for Pearson's correlation) to constrain the bias correction, it seems somewhat inconsistent that Spearman's correlation is used to evaluate how well the various methods preserve the inter-variable "correlation". I understand the reasoning for using nonparametric correlation, but it also raises questions about what the goal of the bias correction should be. That is, should bias correction preserve covariance in instances where covariance can't reliably be estimated?

Specific comments:

1) The article is mostly well written, but there's some awkward phrasing here and there including the Abstract (e.g., "climate variables evolutions", "not well apprehended") and elsewhere ("permits to relate").

2) "methodology" refers to the study of methods. The authors should just use "method" wherever they have "methodology".

3) The last paragraph of the Introduction can be deleted (this structure is obvious).

4) Section 2: "RPC" should be RCP.

5) The framework for the CDF-t method described in text and in Appendix A seems to be a generic accounting of quantile mapping. What I didn't see was information on the transfer function itself (i.e., what criteria determine the degree to which the two distributions are required to match).

6) Table 1 is a helpful summary of the bias-correction methods, but it's hard to match it up with the several pages of text and the Appendices to try to figure out what makes them distinct. There seems to be a gap between describing the general characteristics of the methods (Table 1) and the lengthy text-based descriptions. Please consider adding another table (or other information to Table 1) that helps to determine the specific attributes that makes the methods distinct.

7) At the beginning of Section 4, it would be helpful for the authors to outline what using the three different designs (2d, spatial, and full) aims to accomplish. Which approach has been more commonly used in the literature? Some of the dimensionality tradeoffs could be introduced here as well.

---

## Author Comment (AC1) · 4 May 2020

**Response to Referee Comment 1 on "Multivariate bias corrections of climate simulations: Which benefits for which losses?" by Bastien François et al.**

**Jakob Zscheischler (Referee)**

**Comment:**

This is a timely paper providing an overview about the plethora of newly emerging multivariate bias correction approaches that have been developed over the recent years. The authors provide recommendation about which approach should be used under which conditions. The paper has the potential to become a key reference for multivariate bias correction approaches. It has an easy to follow clear structure, is well written and falls into the scope of ESD. I have a few minor recommendations which should help to improve its accessibility and impact.

**Response:**

We would like to thank Dr. Zscheischler for his very positive comments. We also thank him for the detailed remarks that we will try to include in the updated manuscript. All the comments and our point-by-point responses are given below.

**Comment:**

Introduction: I miss some strong arguments why and in which situations we need MBC. For many impacts, univariate BC is (probably) enough and MBC does not provide are large boost in performance. Indeed a number of studies have argued over the last years that for their application domain MBC does not outperform univariate BC (Yang et al., 2015; Casanueva et al., 2018; Räty et al., 2018). However, I would argue that these results cannot be generalized. One particularly relevant field of application where MBC should be highly beneficial is the area of compound events, where multiple climate drivers result in a large impact (Zscheischler et al., 2018). Arguably, a bias in the dependence structure of the drivers can result in unknown biases of the modelled impacts, which may even be aggravated by univariate BC (Zscheischler et al., 2019).

**Response:** We agree with this comment and propose the following corrections (in blue) in the introduction from L40 of the initially submitted article:

"Although univariate distribution features are adjusted according to references, it can generate inappropriate multivariate situations where the dependence structure between variables and sites is not corrected from the model **and misrepresented** (Maraun, 2013), **or even modified**. Ignoring the observed inter-variable and inter-site dependencies in the correction procedure can result in obtaining corrected outputs with inappropriate physical laws, and thereby distorting the results of impact studies (**Zscheischler et al, 2019**). It is therefore of paramount importance to adjust the dependence structures of climate simulations, in addition to 1d-characteristics, before using it in subsequent studies.

These methodological issues have led up to the recent development of a few multivariate bias correction (MBC) methods. Not only do these methods adjust univariate distribution features, they are also aimed at correcting the dependence structure of climate simulations. Recent studies have shown that univariate BC methods can already provide adequate results for certain specific regional impact studies (Yang et al., 2015; Casanueva et al., 2018), and that using MBC methods does not necessarily present substantial benefits (Räty et al., 2018). However, this does not call into question the interest of MBC methods as these specific results cannot be generalized to each method and application. In particular, MBC methods could be valuable in larger-scale impact modelling frameworks such as compound events, where the combination of physical processes across multiple spatial and temporal scales leads to significant impacts (Zscheischler et al, 2018). As mentioned by Vrac (2018), and

completed by Robin et al. (2019), **MBC methods** may be grouped into three main categories of approaches: the "marginal/dependence" correction approach, the "successive conditional" correction approach, and the "all-in-one" correction approach."

**Comment:**

I'm not sure I entirely agree with the interpretation of Section 5.5.2 and figure 7. As I understand it, Wd only measures a distance. Hence if one obtains a similar value >0 it is unclear whether the change goes into the same direction. One might obtain a similar value for Wd but very different changes in the underlying distributions (though I admit that this would be coincidence and might not be very likely). I think this caveat should be mentioned.

**Response:**

We agree with this comment and want to thank Dr. Zscheischler for this remark. We suggest to add the following sentences (in blue) in the paragraph starting at L489 of sub-section 5.5.2 (Results / Analysis of change in spatial correlations) of the initially submitted article:

"In particular, computing Wd using ranks instead of raw values allows removing the change in the univariate distributions from that in spatial and inter-variable relationships. However, comparing Wd values of climate datasets must be made with caution. Indeed, similar values of Wd for different climate datasets do not necessarily imply that their changes of spatial structure are similar. Results for the three Wasserstein distances on ranks are displayed in Fig. 7 for both France and Brittany. Additional results for Wd on raw values are displayed in Fig. S7 for information purposes only.

For France (Fig. 7a), the three Wd are slightly higher for the reference than for the model data (represented by straight lines). Although the differences are quite small, it cannot be concluded directly that changes of spatial structure are identical, as there is no particular reason for this. For CDF-t outputs, similar Wd are obtained as those from the model. However, as the 1d-BC method does not modify (too much) rank sequence of temperature and precipitation time series, it can be deduced that CDF-t outputs globally reproduce/preserve the spatial structure change of the model."

We also suggest to correct the following paragraph starting at L512 in the same sub-section 5.5.2 (Results / Analysis of change in spatial correlations) of the initially submitted article as follows:

For both dOTC and MBCn outputs, Wd are higher than those from the model. Although the changes in spatial correlations derived by these two methods are too strong, it nevertheless highlights their ability to capture such a change from the model and to use it in their bias correction procedure. Moreover, as explained in subsection 5.4, dOTC and MBCn methods modify only slightly the rank structure of the initial simulations. It can then be deduced that the changes in spatial correlations measured for the two methods are (partially) in agreement with those from the model. However, for MBCn, the three Wasserstein distances increase according to the number of dimensions considered in the bias correction, from 2d- to Full-versions. It can be linked with the deterioration of the quality of results already observed for spatial features for very high-dimensional bias correction. Regarding MRec, and without speaking about its Full-version, similar observations can be made for 2d- and Spatial-outputs as well. In a general way, the Wd associated to the different configurations for dOTC, MBCn and MRec are always above the Wasserstein distances for R2D2, illustrating somehow the assumptions made by these methods about the stationary or non-stationary copula functions.

**Comment:**

L604: Other examples for changes in dependence that might be highly relevant for impacts are: increases in the dependence between storm surge and heavy precipitation in US coasts in the historical period (Wahl et al., 2015): affects the risk of compound floods; - increase in the strength of dependence between seasonal summer temperature and precipitation of most land regions with increasing warming (Zscheischler & Seneviratne, 2017): affects the likelihood of compound hot and dry events with a large array of impacts

**Response:**

To account for this remark, we propose a change of the sentence starting at L604 in the Discussion and recommendations sub-section of the initial submission as following:

"In a general way, copula non-stationarity for future periods can be reasonably expected, e.g. as documented for rainfall spatial distributions (Wasko et al., 2016), for the dependence between storm surge and rainfall (Wahl et al., 2015) and the dependence between seasonal summer temperature and precipitation (Zscheischler & Seneviratne, 2017). However, on the contrary, it can be argued that inter-variable and spatial dependence structures can be assumed to be stable over time for specific regions, because, to some extent, they can be considered as imposed by physical regional constraints (Vrac, 2018)."

**Comment:**

L642: This is easier said than done. The largest challenge in evaluating impact modelling output is the availability of impact data. It will therefore be difficult to decide which BC approach is more appropriate. That said, I agree that creating an ensemble of different approaches might help to cover uncertainties that are not only related to the choice of the GCM and forcing scenario but also the choice of BC method.

**Response:**

We agree with this comment and propose the modification of the sentences starting at L641 (Subsection 6.3/Future Work) as follows:

"Moreover, as mentioned in the introduction section, bias adjusted simulations are particularly valuable for impact studies. **Despite the challenge of missing impact data**, evaluating how the quality of multivariate bias-corrected data influences the results of complex impact models is an important perspective. **Providing** such an analysis will be useful for the scientific community working on climate change impacts, e.g., in hydrology, agronomy or ecology."

**Comment:**

Figure 2 and 4: The correlations could be plotted as difference to the reference to highlight the differences

**Response:**

After careful consideration, we decided to change Figures 2 and 4 as suggested in this comment, by plotting differences with respect to the reference for Spearman correlations (Fig. 2) and order 1 Pearson autocorrelation for temperature (Fig. 4). However, in order to be consistent, we think that these changes imply to change Figure S2 and S4 in the supplementary materials. Captions and numbering of figures will be changed accordingly. Conclusions of the analysis based on Figure 2 and 4 do not change but few modifications of the text are needed to match the updated format of the new figures.

- Figure S2 previously corresponded to relative differences (in %) of Spearman correlations. We now propose to replace previous Fig. S2 by the initial Figure 2 to provide equivalent information to readers.

---

## Author Comment (AC2) · 4 May 2020

**Response to Referee Comment 2 on "Multivariate bias corrections of climate simulations: Which benefits for which losses?" by Bastien François et al.**

**Anonymous Referee #2**

**General comments:**

**Comment:**

1) A comparison of methods is especially helpful in an emerging area of research such as multivariate bias correction (MBC), where few guidelines are available. Overall, I like the article but have some suggestions for improvement.

**Response:**

We would like to thank the anonymous referee for her/his very positive comments and the detailed questions. All the comments and our point-by-point responses are given below.

**Comment:**

2) I was surprised that the authors re-gridded the 0.5 degree precipitation product to the coarser climate model grid using nearest neighbor interpolation. The authors don't say what method is used for the 8km precipitation product, but presumably that also is nearest neighbor? Overall, this approach seems to ignore a lot of spatial information in the "observed" data and effectively makes this a quasi-regular resampling of the observed data and not an interpolation. Is the goal to get area-averaged precipitation or a gridded product of point precipitation? Some discussion of this choice and the tradeoffs is warranted, unless the authors decide to use a different approach.

**Response:**

We think that there is a misunderstanding concerning the re-gridding step. We did not re-grid the 0.5 degree precipitation and temperature products (WFDEI) to the coarser climate model grid (IPSL) as understood in your comment, but the opposite, as explained in Section 2, L94. However, with the aim of clarifying better this point, we propose to rewrite L94 as follows :

"Note that, as spatial resolution between WFDEI and IPSL-CM5 are different, IPSL **model** data are regridded by a nearest neighbour technique to associate each IPSL grid cell to its nearest WFDEI grid cell center. Hence, in the following, the IPSL data will be used at the 0.5° spatial resolution corresponding to that of the WFDEI reference dataset."

Concerning the method used for the 8km precipitation product, nearest neighbour technique is indeed used. To better clarify this, we propose to rewrite L99 in Section 2 as follows:

"IPSL data are regridded to the 8 km x 8 km SAFRAN resolution using nearest neighbour technique".

**Comment:**

3) A lagged version of a variable, whether spatial or temporal, can be thought of as just another variable. Adding a lagged variable to MBC, therefore, is just MBC. As a result, I think the results that show the impacts of poorly conditioned matrices is the key takeaway here. One should parsimoniously add variables that are important to preserving the kind of variability that one is most interested in. For instance, if heatwaves are of interest, then one should emphasize temporal correlation. I'd like to see a bit more on the tradeoffs between emphasizing temporal vs spatial

correlation, in terms of choosing the dimensionality of the bias correction. The authors touch on this in the discussion, but some more specific guidance for making these choices would be helpful.

**Response:**

We agree on the fact that adding variables for MBC must be done wisely by end-users, and propose to add the following sentence in the discussion (in blue), on current L586 in the discussion (Section 6):

"More generally, for most MBCs, for a given number of statistical dimensions (e.g., number of grid cells), as going from a large (e.g., France) to a smaller (e.g., Brittany) area reduces the "effective dimension", it facilitates the multivariate corrections and therefore improves the results (e.g. compare Figs. 1, S1, 4, S4, 5 and S5). This raises the question of whether applying MBC on climate simulations over large geographical areas is justified, i.e. if it is worth striving for the correction of correlation structures between distant sites presenting weak statistical relationships, and, by doing so, taking the risk of losing global effectiveness of the BC methods. It also highlights the importance of choosing parsimoniously the variables to correct, in order to adjust dependence structures that are relevant without potential quality loss induced by additional (and unneeded) variables."

Concerning the tradeoffs between emphasizing temporal vs spatial correlation, this is a relevant remark. However, providing guidance on the compromise between correcting temporal and spatial correlation would require additional evaluations by implementing another dimensional configuration in the study (for-example a 14d-version would be needed to correct autocorrelations of 2 physical variables until lag 7 at one given location). Although useful for end-users interested in correcting temporal correlations, it goes beyond the scope of this paper, and is left for further work.

**Comment:**

4) Given that the methods use covariance (which is the basis for Pearson's correlation) to constrain the bias correction, it seems somewhat inconsistent that Spearman's correlation is used to evaluate how well the various methods preserve the inter-variable"correlation". I understand the reasoning for using nonparametric correlation, but it also raises questions about what the goal of the bias correction should be. That is, should bias correction preserve covariance in instances where covariance can't reliably be estimated?

**Response:**

We disagree with the statement that all the MBC methods presented in the paper use covariance to constraint the correction. Actually, only the "MRec" method, based on a matrix recorrelation technique, uses explicitly covariance/Pearson's correlation to constrain the bias correction. Pearson coefficient measures the strength of the linear relationship between normally distributed variables. Arguably, precipitation is not normally distributed and the relationship between temperature and precipitation can be non-linear. In that sense, to evaluate inter-variable correlations, we think that it is more appropriate to use the Spearman correlation that does not require the assumption of normal distribution of the variables or linear relationship.

Moreover, the Spearman's (rank) correlation is a measure of dependence between 2 variables rid of their marginals. Univariate BC methods are supposed to adjust marginal distributions. Most univariate BC methods (as quantile-quantile like methods) do correct the marginal distributions but leave the dependence (i.e., copula) structure of the model data unchanged. Hence, when applying a multivariate BC, the expectation is to adjust not only the marginals but also the dependence between the variables of interest, regardless of the marginal biases or correctness. The Spearman's correlation is thus appropriate for this specific aspect.

With the aim of justifying better the use of Spearman's correlation for the evaluation of inter-variable dependence structure, we propose to rewrite part of the paragraph starting at L294 in the sub-section 5.2 as follows:

"To evaluate inter-variable dependence structure, Spearman correlations between temperature and precipitation are computed at each grid cell to measure the monotonic relationship between the two physical variables. Using rank correlation presents the particularity of not being value-dependent, i.e. it measures the dependence between two variables rid of their univariate distributions. As the goal when applying MBC is to adjust not only the univariate distributions but also the dependence structure between the variables of interest, Spearman's correlation is appropriate for this latter aspect. Moreover, this measure does not require any assumption on the distribution of the variables or their statistical relationships. It is hence appropriate for temperature and precipitation studies presenting extreme values and/or lower bound (Vrac and Friederichs, 2015)."

**Specific comments:**

**Comment:**

1) The article is mostly well written, but there's some awkward phrasing here and there including the Abstract (e.g., "climate variables evolutions", "not well apprehended") and elsewhere ("permits to relate").

**Response:**

Following this comment, we propose the following corrections (in blue) :

L1 (in the Abstract) "Climate models are the major tools to study the climate system and its evolutions in the future."

L9 (in the Abstract) replacing "not well apprehended yet" by "not yet fully apprehended"

L117 (in Section 3) "permits to relate" by "permits to link"

**Comment:**

2) "methodology" refers to the study of methods. The authors should just use "method" wherever they have "methodology"

**Response:**

The word "method" instead of "methodology" will be used for L13 and L14 in the Abstract.

**Comment:**

3) The last paragraph of the Introduction can be deleted (this structure is obvious).

**Response:**

We prefer to keep the last paragraph of the Introduction. Although the structure of the paper is obvious, it permits to explicitly outline the sections of the paper.

**Comment:**

4) Section 2: "RPC" should be RCP.

**Response:**

Addressed, thanks.

**Comment:**

5) The framework for the CDF-t method described in text and in Appendix A seems to be a generic accounting of quantile mapping. What I didn't see was information on the transfer function itself (i.e., what criteria determine the degree to which the two distributions are required to match).

**Response:**

CDF-t and quantile mapping have indeed a similar philosophy. For the period of calibration, the two methods are theoretically equivalent. However, the difference between CDF-t and quantile mapping lies in the correction of variables during the period of projection. CDF-t takes into account the change in the modelled CDFs from the calibration to the projection period, while quantile mapping projects directly the modelled values onto the CDF of the calibration period to compute quantiles.

More specifically, Appendix A does not correspond at all to a generic quantile-mapping. Equations (A1) to (A4) describe the construction of a transfer function allowing to go from the reference CDF  $F_{RC}$  (i.e., over the calibration time period) to an estimate of a projected pseudo-reference CDF  $F_{RP}$  (i.e., over the projection time period). This projected pseudo-reference CDF  $F_{RP}$  is then used in a second step, involving quantile-mapping. While a traditional quantile-mapping approach performed to correct a dataset  $X_{Mp}$  of simulations over the projection period will use the formulation  $\hat{X}_{Mp} = F_{Rc}^{-1}(F_{Mc}(X_{Mp}))$  to get a corrected value  $\hat{X}_{Mp}$  (i.e., based on 2 distributions characterizing the calibration period), the CDF-t method relies on the following formulation:  $\hat{X}_{Mp} = F_{Rp}^{-1}(F_{Mp}(X_{Mp}))$ , where the 2 involved distributions characterize projected distributions. Those points are already mentioned in Appendix A. However, with the aim of clarifying this point, we propose to rewrite the paragraph starting at L690 of Appendix A as following:

"Once  $F_{Rp}$  has been estimated, a simple quantile-quantile method is performed between  $F_{Rp}$  and  $F_{Mp}$  to derive the bias corrected time series of CDFs  $\hat{X}_{Mp}$  for the projection period as following:

$$\widehat{X}^{d}_{M_{p}}(t) = F_{Rp}^{-1}(F_{Mp}(X^{d}_{M_{p}}(t)))$$

(A5)

While a traditional quantile-mapping approach performed to correct a dataset  $X_{Mp}$  of simulations over the projection period will use the formulation  $\hat{X}_{Mp}^{d}(t) = F_{Rc}^{-1}(F_{Mc}(X_{Mp}^{d}(t)))$ , (i.e., based on two distributions characterizing the calibration period), the CDF-t method relies on Eq. (A5) where the two involved distributions characterize projected distributions. By proceeding this way, CDF-t takes into account the potential evolution of CDFs of the model between the calibration and projection periods to adjust the projection period. CDF-t is applied independently for each of the *D* statistical dimensions and for both calibration and projection period to derive the final bias corrected outputs  $\hat{X}_{Mc}$  and  $\hat{X}_{Mp}$ .

**Comment:**

6) Table 1 is a helpful summary of the bias-correction methods, but it's hard to match it up with the several pages of text and the Appendices to try to figure out what makes them distinct. There seems to be a gap between describing the general characteristics of the methods (Table 1) and the lengthy text-based descriptions. Please consider adding another table (or other information to Table 1) that helps to determine the specific attributes that makes the methods distinct.

**Response:** Agree. Thanks for this suggestion. In order to provide a helpful summary of the methods, we propose to add another table in Section 3 (Multivariate bias correction methods):

| Characteristics             | CDF-t                                  | R2D2                            | dOTC                                        | MBCn                                         | MRec                                                        |
|-----------------------------|----------------------------------------|---------------------------------|---------------------------------------------|----------------------------------------------|-------------------------------------------------------------|
| Type of BC                  | 1d-BC                                  | МВС                             | МВС                                         | МВС                                          | МВС                                                         |
| Category of
MBC approach | n.a.                                   | Marginal/depe
ndence         | All-in-one                                  | Marginal/depen
dence                      | All-in-one                                                  |
| Statistical
technique    | Non-stationar
y quantile
mapping | Conditionnal resampling         | Optimal
transport                        | Iterative partial
matrix
recorrelation | Matrix
recorrelation                                     |
| Dependence
structure     | ~ same as the
model                 | ~ same as the reference         | Allows
changes in
the dep.
struct. | Allows
changes in the
dep. struct.     | Allows
changes in the
Gaussian dep.
struct. |
| Conceptual
feature       | Deterministic                          | Deterministic
and stochastic | Stochastic                                  | Deterministic
and stochastic              | Deterministic                                               |

| Table 1. Summary | of attributes   | of the differe | ent bias-correction | methods |
|------------------|-----------------|----------------|---------------------|---------|
|                  | y of allindated |                |                     | methods |

We also propose to modify the paragraph starting at L109 (Section 3) as follows:

"This section presents a brief description of the univariate BC method and the four multivariate BC methods implemented in this study. As a reminder, results from the univariate CDF-t method serve as a benchmark to measure the benefits of considering multivariate aspects in the correction procedure instead of using univariate BC methods. For sake of clarity, Table 1 provides a concise summary of the different attributes that make the BC methods distinct."

Numbering of tables will be changed accordingly.

**Comment:**

7) At the beginning of Section 4, it would be helpful for the authors to outline what using the three different designs (2d, spatial, and full) aims to accomplish. Which approach has been more commonly used in the literature? Some of the dimensionality tradeoffs could be introduced here as well.

**Response:**

The information concerning the aim of each dimensional design (2d, Spatial and Full) is already formulated at the beginning of Section 4 in the form of bullet points (starting at L210 in Section 4 - Design of Experiments). However, we propose the following clarifications (in blue) at L206 (Section 4 - Design of Experiments):

« We tested and assessed this approach for each method, but also expanded the study to include high-dimensional configurations of MBC to adjust spatial and full (i.e. spatial and inter-variable jointly) dependence structures of climate simulations. **Depending on the dimensional configurations, the objectives of corrections for multivariate properties differ. Including different dimensional**  versions in the study will permit to better highlight the potential losses and benefits associated with them. Therefore, in the following, each of the four MBC methods is applied according to the three following configurations: »

Concerning the most commonly used approach, in our opinion, L204 of Section 4 is quite clear: It indicates that, in most cases, inter-variable configurations (i.e. in our study referred to as 2d-version) are applied in the literature and two papers are cited (Meyer et al., 2019; Guo et al., 2019) to illustrate this point. Of course, as shown by the results later in the article, this choice of inter-variable configuration can have important consequences on spatial and temporal dependencies.